# Natural image synthesis for the retina with variational information bottleneck representation

**Babak Rahmani**

rahmani.b91@gmail.com

**Demetri Psaltis**
EPFL
demetri.psaltis@epfl.ch

**Christophe Moser**
EPFL
christophe.moser@epfl.ch

## Abstract

In the early visual system, high dimensional natural stimuli are encoded into the trains of neuronal spikes that transmit the information to the brain to produce perception. However, is all the visual scene information required to explain the neuronal responses? In this work, we search for answers to this question by developing a joint model of the natural visual input and neuronal responses using the Information Bottleneck (IB) framework that can represent features of the input data into a few latent variables that play a role in the prediction of the outputs. The correlations between data samples acquired from published experiments on ex-vivo retinas are accounted for in the model by a Gaussian Process (GP) prior. The proposed IB-GP model performs competitively to the state-of-the-art feedforward convolutional networks in predicting spike responses to natural stimuli. Finally, the IB-GP model is used in a closed-loop iterative process to obtain reduced-complexity inputs that elicit responses as elicited by the original stimuli. We found three properties of the retina's IB-GP model. First, the reconstructed stimuli from the latent variables show robustness in spike prediction across models. Second, surprisingly the dynamics of the high-dimensional stimuli and RGCs' responses are very well represented in the embeddings of the IB-GP model. Third, the minimum stimuli consist of different patterns: Gabor-type locally high-frequency filters, on- and off-center Gaussians, or a mixture of both. Overall, this work demonstrates that the IB-GP model provides a principled approach for joint learning of the stimuli and retina codes, capturing dynamics of the stimuli-RGCs in the latent space which could help better understand the computation of the early visual system.

## 1 Introduction

A fundamental challenge in neuroprosthetics is finding the proper input to a sensory or motor system that yields a desired functional output. This is achieved naturally by the underlying physiological circuit in an unimpaired system. For example, in the vision system, when an image is formed onto the photoreceptors, the electrical neural activity is processed by several layers of neurons within the retina, and a train of electrical neural spikes is sent to the visual cortex via the optic nerve producing a perception of the visual scene. However, one can ask: is it possible to reduce the complexity of the input stimuli and still obtain the same functional behavior? In other words, what are the reduced spatiotemporal stimuli that elicit the same responses as those of the higher-dimensional original inputs? These are important questions both for understanding the neural computations of the retina and possibly for designing visual prosthesis in which minimal information stimuli that optimally drive neurons may prove beneficial [36, 18, 22].

In this work, we focus on the *natural* vision system and search for answers to these questions. Our system consists of the spatiotemporal stimuli impinging on the photoreceptors in the retina up to the spiking responses emitted by the Retinal Ganglion Cells (RGCs). Linear-nonlinear (LN) [11] and

36th Conference on Neural Information Processing Systems (NeurIPS 2022).

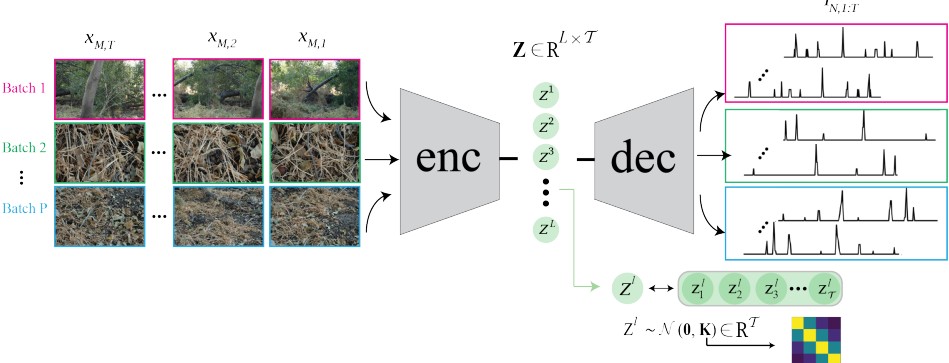

Figure 1: Overview of the IB-GP model of the retina responses. Batches of $T$ high-dimensional stimuli are encoded into a low-dimensional space and decoded back to the batches of $N$ neuronal response trains of length $T$. The latent space consists of $L$ latent vectors. Each latent vector $z^l$ is modeled by a GP prior to account for the sample correlations in the original space.

generalized linear models (GLMs) [25, 26] were proposed for encoding of the retinal neural codes. Although successful for artificial stimuli such as white noise, they could not faithfully describe RGCs' responses to natural stimuli [15]. The convolutional neural network (CNN) model of McIntosh et al. [21] was the first model capable of such computation. This was followed by other efforts such as employing recurrent neural networks [5] or variants of CNNs [20]. Although high-dimensional in nature, neural activity can be well explained by low-dimensional representations [12, 14, 28]. This opens up the possibility of using models that learn a latent representation of data, such as VAEs [16] and ICAs [19]. Accordingly, latent dynamical methods that inspect sources of low-dimensional structure in the neural response have recently gained growing interest [17, 35].

Inspired by the Deep Variational Information Bottleneck (DVIB) [2, 9], we introduce the Information Bottleneck Gaussian Process (IB-GP), a latent space model that can extract the principal features of the *joint* stimuli-responses distribution that are sufficient for predicting the retina responses. The schematic of the IB-GP is depicted in Fig. 1. This model is a variational approximation to the original Information Bottleneck framework of Tishby et al. [31]. The objective of IB is to obtain a reduced representation of the input source that preserves maximum information about the output response. DVIB assumes the IB's latent codes are i.i.d. This is a poor assumption for the retina dataset. The reason for this is two-fold. First, while observing a scene, images captured by the retina are naturally very similar. Second, during fixation, the eyes undergo dynamic movements that shift the gaze's center. Therefore, incorporating the temporal correlations among data samples should provide a better model. Accordingly, IB-GP uses a Gaussian Process prior to modeling the retina dataset.

Our contributions to this work are:

- **A probabilistic model**: We propose IB-GP, a new model for the RGC's spike train in response to the complex natural visual stimuli using the IB framework. The model integrates a GP prior into the IB latent code that can learn the temporal dynamics of the data in a lower dimension space. This model shows competitive performance compared to the state-of-the-art feedforward CNN models of the retina tested on real-world experimental data.

- **Latent dynamics analysis**: We analyze the latent space of the proposed model and extract low-dimensional dynamics consistent with the high-dimensional data dynamics. We show that the extracted dynamics are more predictive of the neural responses compared to dynamics extracted by a model with no temporal prior constraint.

- **Closed loop image synthesis**: Utilizing the IB-GP model, which only allows principal features of the input stimuli to pass the bottleneck, the original input stimuli are pruned to contain the minimum information required for producing the RGC responses. The resulting complexity-reduced stimuli could be used for the next round of measurements in an iterative process.

In the following, we start by reviewing the related work in section 2, the general setting of the problem and our model are presented in section 3, and the closed-loop stimuli optimization procedure is outlined in section 4. Experiments and results are presented in sections 5 and 6[1]. Finally, the results are discussed in section 7.

## 2 Related work

In the RGC modeling literature, our work is the closest to McIntosh et al. [21], in which a CNN is used to model the responses of RGCs to a set of input stimuli. Despite the simplicity of this approach, the feedforward CNNs for image classification or regression cannot be directly utilized to extract the principal features of the stimuli contributing to the neural responses. On the other hand, latent neural models such as [17, 23, 29] are primarily applied to cortical recordings to learn sources of neural variability in the motor cortex data. Unlike our model, which directly relates the stimuli to the neural responses, the stimuli in those lines of work are utilized as a guide star to disentangle the latent factors. Therefore, these methods do not directly apply to modeling the RGC response to input stimuli. In the closed-loop physical system input optimization literature, our work is related to [27, 30, 34, 4]. In particular, for neuroscience applications, authors in [4] control the activity of the individual neuronal sites in V4 by optimizing individual input stimulation, whereas in our work, thanks to the learned latent code of the entire population of the targeted neurons, all stimuli are optimized together. In Shah et al. [30], the RGCs' responses to electrical stimulation were optimized by first developing a model for the electrical stimuli and the spiking probabilities and then using the model for adaptive stimulation. The model is obtained by parameterizing the spike amplitude and electrical stimulation threshold by a few parameters and then maximizing an evidence lower bound on the spiking probabilities. Our work assumes no relevant relation between the input stimuli and the spiking probabilities. Instead, the relevant parts of the input for predicting the output are discovered automatically.

## 3 Methods

Below, we introduce the IB-GP method, which models the neural activities by learning the latent space underlying the stimuli and RGCs' responses. We also summarize additional methods, including a variant of our proposed method that we used for evaluating the IB-GP model.

### 3.1 IB-GP model

**Notation and problem formulation**  The input stimuli dataset $\mathbf{X} \in \mathbb{R}^{M \times T}$ consists of $T$ consecutive spatial inputs of size $M$, i.e. $\mathbf{X} = [x_{M,1}, x_{M,2}, \cdots, x_{M,T}]$, that are projected in the retina at a constant rate to elicit count responses of the form $\mathbf{Y} \in \mathbb{N}^{N \times T}$, where $\mathbf{Y} = [y_{N,1}, y_{N,2}, \cdots, y_{N,T}]$ and $N$ denotes the number of RGCs. With the above assumptions, we formulate the problem as finding the latent variables $\mathbf{Z} \in \mathbb{R}^{L \times \mathcal{T}}$ that have maximal mutual information $I(\mathbf{Z}, \mathbf{Y})$ with targets $\mathbf{Y}$. We note that $\mathbf{Z}$ is comprised of $\mathcal{T}$ consecutive data points, i.e. $\mathbf{Z} = [z_{L,1}, z_{L,2}, \cdots, z_{L,\mathcal{T}}]$ where $\mathcal{T} \leq T$. Also $\mathbf{Z}$ is simultaneously constrained to have minimal mutual information with $\mathbf{X}$. Therefore, the constraint optimization problem can be written as

$$\mathcal{L}_{IB} = \max_{\zeta} \big[ I_{IB} \big]$$

where

$$I_{IB} = I(\mathbf{Z}, \mathbf{Y}; \zeta) - \beta I(\mathbf{Z}, \mathbf{X}; \zeta) . \tag{1}$$

$\zeta$ designates the parameters of the model, i.e. $\theta$ and $\phi$, (notation henceforth dropped for brevity) and $\beta$ is a variable to adjust the amount of reduced and preserved information in $\mathbf{Z}$. To obtain these latent variables, we approximate their posterior distribution with a parametric stochastic encoder $p_\phi(\mathbf{Z}|\mathbf{X})$. Due to the *spatio-temporal* nature of the data, assumption of data independence along the time dimension is not valid. The same is true in the latent space. To account for this fact, we assume a posterior of the form:

$$p_\phi(\mathbf{Z}|\mathbf{X}) = \prod_{l=1}^{L} \mathcal{N}(\mathbf{z}^l | \boldsymbol{\mu}_\phi^l(\mathbf{X}), \boldsymbol{\Sigma}_\phi^l(\mathbf{X})) \tag{2}$$

---

[1]Code for reproducing the results are provided in Appendix 9.

where $\mathbf{z}^l$ denotes the $l$-th latent vector of size $\mathcal{T}$. In Eq. 2, $\boldsymbol{\mu}_\phi^l \in \mathbb{R}^\mathcal{T}$ and $\boldsymbol{\Sigma}_\phi^l \in \mathbb{R}^{\mathcal{T} \times \mathcal{T}}$ are the mean and covariance matrix of a multi-variate normal distribution that are functions of the inputs. We choose the structure of the covariance of the multivariate normal distribution in Eq. 2 to reflect the time correlations in the data. Hence, similar to previous work [3, 7, 13], we construct the $\boldsymbol{\Sigma}_\phi^l$ in our model by the product of bidiagonal matrices:

$$[\boldsymbol{\Sigma}_\phi^l]^{-1} = \mathbf{V}_l^\mathrm{T}\mathbf{V}_l + \mathbf{I} \text{ where } [\mathbf{V}_l]_{\tau\tau\prime} = \begin{cases} v_{\tau\tau\prime}^l & \tau\prime \in \{\tau, \tau+1\} \\ 0 & \text{o.w.} \end{cases} \tag{3}$$

We note that the simple GP kernel assumption makes the computation required for drawing samples from $p_\phi$ linear in time [3]. Expanding the IB objective in Eq. 1 (details in the Appendix 1), we have:

$$I_{IB} = \int d\mathbf{Y}\,d\mathbf{Z}\,p(\mathbf{Y},\mathbf{Z})\,\log \frac{p(\mathbf{Y}|\mathbf{Z})}{p(\mathbf{Y})} - \beta \int d\mathbf{X}\,d\mathbf{Z}\,p(\mathbf{X},\mathbf{Z})\,\log \frac{p_\phi(\mathbf{Z}|\mathbf{X})}{p(\mathbf{Z})} \tag{4}$$

Although all terms in the RHS of Eq. 4 are fully defined, computing marginal distributions $p(\mathbf{Z})$ and $p(\mathbf{Y}|\mathbf{Z})$ may be intractable. We use a variational approximation for $p(\mathbf{Y}|\mathbf{Z})$ that is parameterized with $\theta$. On the other hand, the prior on the latent variables, i.e. $p(\mathbf{Z})$, is modeled using GPs. In particular, we assume a GP prior on $\mathbf{Z}$ defined as a multivariate normal:

$$\rho(\mathbf{Z}) = \prod_{l=1}^L \mathcal{N}(\mathbf{z}^l|\mathbf{0}, \mathbf{K}) \tag{5}$$

where $\rho$ is the variational approximation for the prior and $\mathbf{K}$ is the covariance function that models the temporal correlations in the latent space. In more details, the covariance between the $\tau$-th and the $\tau\prime$-th samples is computed as $\mathbf{K}_{\tau\tau\prime} = \mathcal{K}(\tau, \tau\prime)$ where $\mathcal{K}$ is the kernel function. We used the Cauchy kernel:

$$\mathcal{K}(\tau, \tau\prime) = \sigma^2 \big(1 + \frac{(\tau - \tau\prime)^2}{l^2}\big)^{-1} \tag{6}$$

where $\sigma$ and $l$ are the magnitude and temporal scales of the kernel function, respectively. Cauchy kernels have proved successful in modeling of the time-series systems with multi-scale dynamics as shown before [13]. Substituting the variational approximations of the intractable marginals in Eq. 4 and using the fact that the Kullback Leibler (KL) divergence is always positive, we obtain a lower bound for the IB objective:

$$I_{IB} \ge \frac{1}{T}\sum_{t=1}^T \left[ \mathbb{E}_{p_\phi(\mathbf{Z}|x_{1:\tau(t)})}[\log q_\theta(y_t|\mathbf{Z})] - \beta D_{KL}[p_\phi(\mathbf{Z}|x_{1:\tau(t)})||\rho(\mathbf{Z})] \right]. \tag{7}$$

where $\tau(t)$ is to represent time dependence up to time $t$. See Appendix 1 for more details.

### 3.2 Other models considered

**Maximum likelihood feedforward CNN** is used as a comparison baseline. We use the same CNN network that was shown in the previous work to obtain state-of-the-art results on a larger version of the *Natural* dataset [21]. The predicted outputs of the network are the averaged maximum likelihood estimation of the retina responses given the input stimuli. Artificial noise is injected into the model during optimization to account for the variability in retinal spiking.

**IB-Disjoint** optimizes the objective from Eq. 7 using the same training procedure as the original method. The difference lies in that the IB-Disjoint assumes latent samples are temporally independent. Accordingly, an isometric Gaussian distribution is employed as the prior in Eq. 7. This model is akin to the standard Variational Auto-encoder [16], or vanilla DVIB [2] in which samples in the data and latent space are assumed to be independent (no latent GP).

## 4 Closed loop stimulation

In this section, we take advantage of the model developed in the previous sections to devise an algorithm that uses the prior recorded data to optimize the stimulation by iteration in subsequent

measurements in a closed loop. Specifically, we assume the latent variables of the learned model, henceforth referred to as the *forward model*, have captured the principal rules governing the underlying biological system. We intend to find a transformed version of the original input stimuli that yields a set of latent variables that produce the most correlated responses with the original target responses in the subsequent round of measurements. Of course, the original stimuli themselves are one set of solutions. However, this is not a useful transformation. Instead, we are interested in obtaining the best transformation subject to a constraint on their complexity. Hence, we define a parametric function $f_\xi : \mathbb{R}^{M \times T} \to \mathbb{R}^{M \times T}$ that maps the complex original stimuli $\mathbf{X}$ onto the transformed stimuli $\mathbf{X}^*$. Passing $\mathbf{X}^*$ to the forward model, the parameter of the mapping functions, i.e. $\xi$, are optimized so that the forward model's output responses $\mathbf{Y}^*$ are the most correlated with the original responses $\mathbf{Y}$. Denoting the original stimuli and responses as $\mathbf{X}^{\text{orig}} \coloneqq \mathbf{X}$ and $\mathbf{Y}^{\text{orig}} \coloneqq \mathbf{Y}$, the objective function reads as:

$$\min_{\xi} \; D_1(\mathbf{Y}^{\text{orig}}, \mathbf{Y}^*) + \alpha \big[ D_2(\mathbf{X}^{\text{orig}}, \mathbf{X}^*) + D_3(\mathbf{X}^*) \big]$$

where:

$$\mathbf{X}^* = f_\xi(\mathbf{X}^{\text{orig}}), \; \mathbf{Z}^* \sim p_\phi(.|\mathbf{X}^*), \; \mathbf{Y}^* \sim q_\theta(.|\mathbf{Z}^*). \tag{8}$$

Therein, $p_\phi$ and $q_\theta$ are the encoder and decoder of the forward model trained on the prior data. $D_1$ is a measure to ensure the closeness of the original target outputs and the responses of the forward model to $\mathbf{X}^*$. $D_2$ is a similarity measure between the original and synthesized stimuli $\mathbf{X}^{\text{orig}}$ and $\mathbf{X}^*$ which encourages the image synthesizer to focus on finding the essential features in the stimuli rather than synthesizing entirely new solutions. Finally, $D_3$ is used to constrain the synthesized stimuli $\mathbf{X}^*$ to be smooth. Contributions of $D_2$ and $D_3$ to the total loss in Eq. 8 are tuned by hyperparameter $\alpha$. Two major mechanisms help to prevent $f_\xi$ from being simply an identity mapping. One, during the training of the $f_\xi$, the encoder's latent variables $\mathbf{Z}^*$ are masked by zeroing out all the latent variables except for the most informative one. This way, $\mathbf{X}^{\text{orig}}$ is no longer the optimal solution that minimizes the first term in the objective function. Two, $f_\xi$ is implemented using an Autoencoder architecture with a limited latent capacity. Finally, the optimized stimuli for subsequent rounds of measurements can be obtained by the following procedure: (1) map the original stimuli to the optimized stimuli using mapping $f_\xi$, (2) encode the new input $\mathbf{X}^*$ to the posterior $\mathbf{Z}^*$, (3) decode the latent variables, (4) construct the metric $D$ and backpropagate the error to fit $f_\xi$, (5) once converged, use the mapping function to obtain the optimized stimuli $\mathbf{X}^*$ (6) send new inputs $\mathbf{X}^*$ to the true biological system (or its model) and observe the new true system outputs, (7) to repeat the procedure, use the true system's new inputs and outputs to fine-tune the forward model and (8) repeat.

## 5  Experiments

### 5.1  Dataset

We focus on applying our model to the time-series dataset containing the 2D train of input stimuli (images) entering the vision circuit of an example subject and the corresponding elicited count responses of a group of neurons. Specifically, we use two real-world experimental datasets. The first one, *Natural* dataset, contains natural scene images as input stimuli and spike trains from nine RGCs in Salamander [21] [2]. The second dataset, *Brownian-movement*, is a public dataset which contains images with disc-like shapes having Brownian movements and the spike trains elicited in 91 Rat's RGCs [8].

### 5.2  Forward modeling

**Training specifications**  We used the models outlined in section 3 to fit the neural activities and input stimuli. The Natural dataset is from the Natural images from the birthplace of the human [32]. This dataset consists of Natural images in the wild that do not belong to any particular category. The train set contains 3950 unique images (each unique image is jittered 100 times) of which 20% was randomly chosen for validation. The test set contains 50 averaged repeated trials of novel stimuli

---

[2]The Natural dataset, used in this work, is a subset of the original dataset of [21] that was provided by the authors therein. This dataset contained responses of only nine RGCs.

where each unique image is jittered 100 times. One batch of the test data consists of 10 unique images. We note that the output responses were first binned at 10 ms and then smoothed using a Gaussian filter with a standard deviation equal to the bin size. The Brownian-movement dataset contains 129600 pairs of input images and output spikes binned at 12.5 ms in the training set, of which 20% was randomly chosen for validation. The test set contains 1800 novel stimuli. We varied the number of neurons in the Brownian dataset for which we tested our model. We report the results for nine neurons in the main paper and 15, 21, 27, 36 and 51 neurons in Appendix 5.

**Models and evaluation metrics**    To account for the discrete nature of the output responses (count time series), we employ a Poisson regression model for both IB-GP and IB-Disjoint decoder as well as the feedforward CNN loss term, i.e. $q_\theta(y_t|\mathbf{Z}) = \text{Po}(y_t|f_\theta(\mathbf{Z}))$ where $f_\theta$ is the parametric network. We sweep the number of latent factors and the prior pressure constraint $\beta$ for both IB-GP and IB-Disjoint. For the Natural and Brownian datasets, the number of latents is swept between 1 to 15 (step 1) and 1 to 50 (step 5) variables, respectively. Except for the latent space of the IB-GP, which requires more variables to model its covariance matrix, IB-GP and IB-Disjoint are implemented using the same architectures. All hyperparameters in the training of the two models are fixed to be the same. We perform a three-fold fitting of the models for the spike prediction results. We report the models' Poisson negative log-likelihoods on the ground truth data. Moreover, we use the Pearson correlation [6] between the ground truth data and the networks' prediction as another metric for evaluating the performance of the models. The architectures of the networks and the training setup are explained in more details in Appendix 2.

**Traversal analysis**    We perform traversal analysis on the latent space of IB-GP and IB-Disjoint models to study the amount of information picked by each latent variable. To this end, the encoder $p_\phi(\mathbf{Z}|\mathbf{X})$ is fed with a batch of input stimuli to initialize the latents. In the IB-GP model, the $l$-th latent is sampled from $\mathcal{N}(\mathbf{0}, m\mathbf{K})$ where $m$ is a multiplier that is traversed in the range $[-3, 3]$ while keeping all the other latents to their inferred values. The latent variables in the IB-Disjoint model are sampled similarly but from the normal univariate distribution $\mathcal{N}(\mathbf{0}, m\mathbf{I})$. The latents are then passed to the decoder that reconstructs the output responses. The output responses of the traversed model are then compared against the non-traversed responses.

### 5.3    Closed-loop image synthesis

We use both the IB-GP and IB-Disjoint with a latent size of 4. The prior constraint parameter $\beta$ is set to 0.05. These hyperparameters are chosen as a trade-off between the latent space compression and RGCs' neural activity fitting accuracy. The image synthesizer $f_\xi$ is implemented using an Autoencoder architecture [33] with a latent size of 15. The rationale for this choice of architecture is explained in Appendix 2. To allow for multi-phase iterative optimization, we use a neural network as the proxy for the true biological system trained on the Natural dataset. This model is trained once and kept fixed. We use the feedforward CNN network of McIntosh et al. [21] as the true model in order to rule out the possibility of obtaining correlated responses due to the similarity of the IB-GP model and the true model. We optimize the stimuli in a three-phase closed-loop experiment. We used the Poisson regression loss as the $D_1$ measure in the objective function 8. The similarity measure $D_2$ between the synthesized and original stimuli is computed using a Mean Squared Error (MSE). Total Variation (TV) metric [10] is used as the $D_3$ measure. We analyze the contributions of all measures to the quality of the synthesized stimuli by sweeping the hyperparameter $\alpha$.

## 6    Results

### 6.1    Forward modeling

**RGC neural activity fitting**    Performances of the IB-GP, IB-Disjoint, and Feedforward CNN in fittings to neural activities versus the number of latents for both Natural and Brownian datasets are plotted in Figure 2. Figure 2a,b shows that IB-GP outperforms IB-Disjoint almost in all cases. For a fixed $\beta$ value, the performance of the IB-Disjoint is usually closer to that of the IB-GP with the increase in the number of latents. This indicates that IB-GP learns neural activities with a fewer number of latents. Similar is the case with $\beta$ value decreasing to the extent where IB-Disjoint's performance reaches that of the IB-GP. For less constrained models, IB-Disjoint surpasses IB-GP

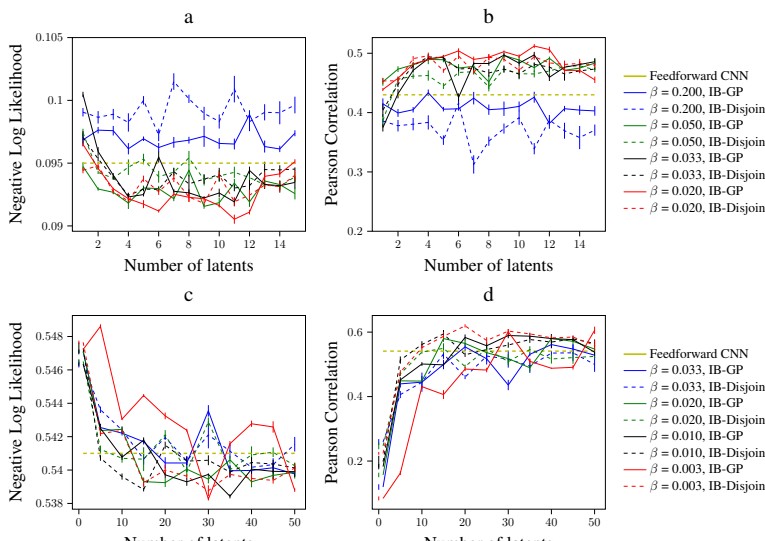

Figure 2: Overview of the RGCs' neural activity prediction accuracy for the IB-GP, IB-Disjoint, and Feedforward CNN trained on the Natural and Brownian dataset. (a) Poisson negative log-likelihood of the predicted neural responses and (b) Pearson correlation of the predicted and true spike responses as a function of the number of latents for different values of the prior pressure constraint $\beta$ for the Natural dataset. (c, d) the same as (a,b) but for the Brownian dataset.

(refer to Appendix 4). Interestingly, IB-GP achieves good neural prediction performance for the Natural dataset even with two latents. Figure 2c,d shows that for the Brownian dataset and larger $\beta$s, IB-GP keeps its performance gap with IB-Disjoint for a large number of latents. This is due to the strong correlation between the Brownian stimuli dataset (refer to Appendix 3 to observe some examples of stimuli). Both IB-GP and IB-Disjoint outperform the Feedforward CNN when a sufficient number of latents are used ($\sim$ 3 latents and $\beta < 0.033$) for Natural and ($\sim$ 10 latents and $\beta < 0.01$) for the Brownian dataset. [3]

**Comparison of the IB-GP and IB-Disjoint learned dynamics** In Figure 3, we analyze the learned dynamics of the IB-GP and IB-Disjoint trained under the same conditions on the Natural dataset (results for the Brownian dataset are in Appendix 5). For $\beta = 0.05$ and 15 latents, performances of the IB-GP and IB-Disjoint in prediction of RGCs' responses are similar (correlation $\sim$ 0.45 vs. $\sim$ 0.43, respectively). Nevertheless, the amount of information picked by latents in each model varies significantly. To show this, we perform a traversal analysis of the latents and report the variability in the output responses for each traversed latent in Figure 3(a,d). The KL divergence for each latent is also plotted. As observed in the figure, IB-GP lumps most of the information required to explain the neural responses in one factor. In comparison, this information is spread over five factors in IB-Disjoint. Analyzing the dynamics of the latents in each model can provide further insight. Figure 3(b,e) visualizes the inferred dynamics of the most informative latent in each model after reducing the dimension to 2 with T-SNE. As observed in the figure, the dynamics of the most informative latent in the IB-GP have 50 clusters. Strikingly, these 50 clusters correspond to the 50 unique stimuli in the dataset. The test set consists of 5 batches in which each batch has 10 unique images (each jittered 100 times). Each batch is colored differently in the figure. Auto-Correlation (AC) plots of the latents corroborate these findings. As evident from Figure 3(c,f), AC of the most informative latent in the IB-GP closely follows the statistics of the true and predicted responses of the model (averaged over all neurons). While the AC of the IB-Disjoint's predicted responses is similar to that of the true responses, the AC of its latents is very different. To demonstrate the efficacy of the IB-GP model in fitting the RGCs' responses, neural activity predictions of both models are shown in Figure 3 (g,h). Refer to Appendix 6 for a more detailed interpretation of IB-GP's latent dynamics.

---

[3]We should emphasize that the hyperparameters of all three networks were not optimized to achieve the best results. Proposing a retina model that outperforms all the other methods is not the main focus of the work; for that requires much further experimentation.

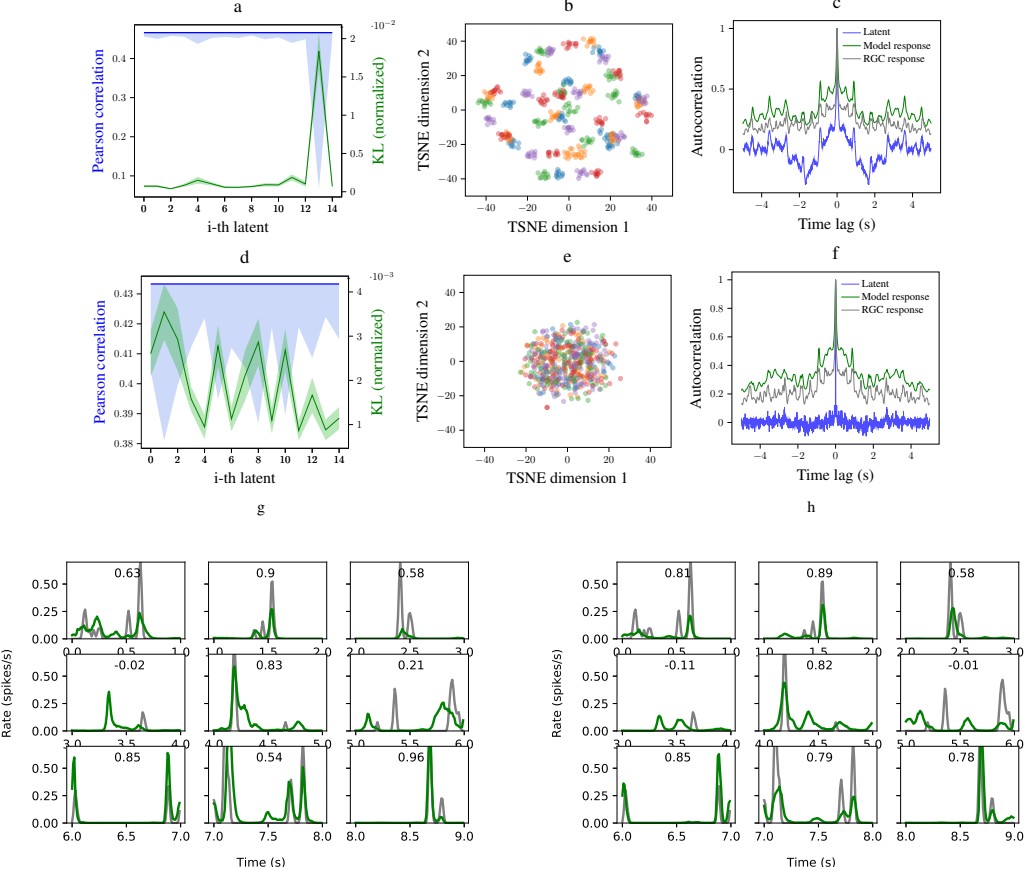

Figure 3: Analysis of the IB-GP (first row) and IB-Disjoint's (second row) learned dynamics evaluated on the Natural images in the test dataset. Third row plots examples of the RGCs' neural responses obtained by the IB-GP (left) and IB-Disjoint (right). (a,d) The average prediction accuracy of the IB-GP and IB-Disjoint, measured by the Pearson correlation, is shown as a flat line. The variability in the accuracy of the model for traversed latent factors is depicted as blue shades. Averaged normalized KL divergence of each latent is also plotted. (b,e) The temporal information of the most informative latent variable from (a,d) are visualized after reducing the dimensions to 2 with T-SNE. Each color denotes one batch of the test set. (c,f) AC of the most informative latent is compared against those of the true temporal activity and the model's predicted responses averaged over the number of neurons. (g,h) Neural activity prediction of the models for some example cells. Numbers show the correlation of the predicted responses with the true activities.

## 6.2 Closed loop image synthesis

Figure 4 reports the training results of the closed-loop experiment after the first round of measurements. Both the IB-GP and IB-Disjoint are used as the forward model in the training of the image synthesizer. We consider two scenarios when tuning the hyperparameters in the objective function of the image synthesizer. Scenario one: the image synthesizer is trained only to reduce the complexity of the original stimuli independently from the RGCs' responses (dropping the $D_1$ term in the objective function of the image synthesizer). In this scenario, we also consider the case where the TV smoothness constraint is dropped. These are important cases because they answer this valid question: can a simple dimensionality-reduction technique be used for synthesizing images that elicit responses correlated with the original ones? Scenario Two: the image synthesizer is trained in conjunction with RGCs' responses ($\alpha \neq 0$). Figure 4(a, b) plot the Pearson correlation and negative log-likelihood of the RGCs' responses predicted by the true model in response to the stimuli obtained by the image synthesizer network for the two scenarios. The TV loss of the obtained synthesized stimuli in each case is plotted in Figure 4c. As can be observed in the Figure 4d, dimensionality reduction of the stimuli

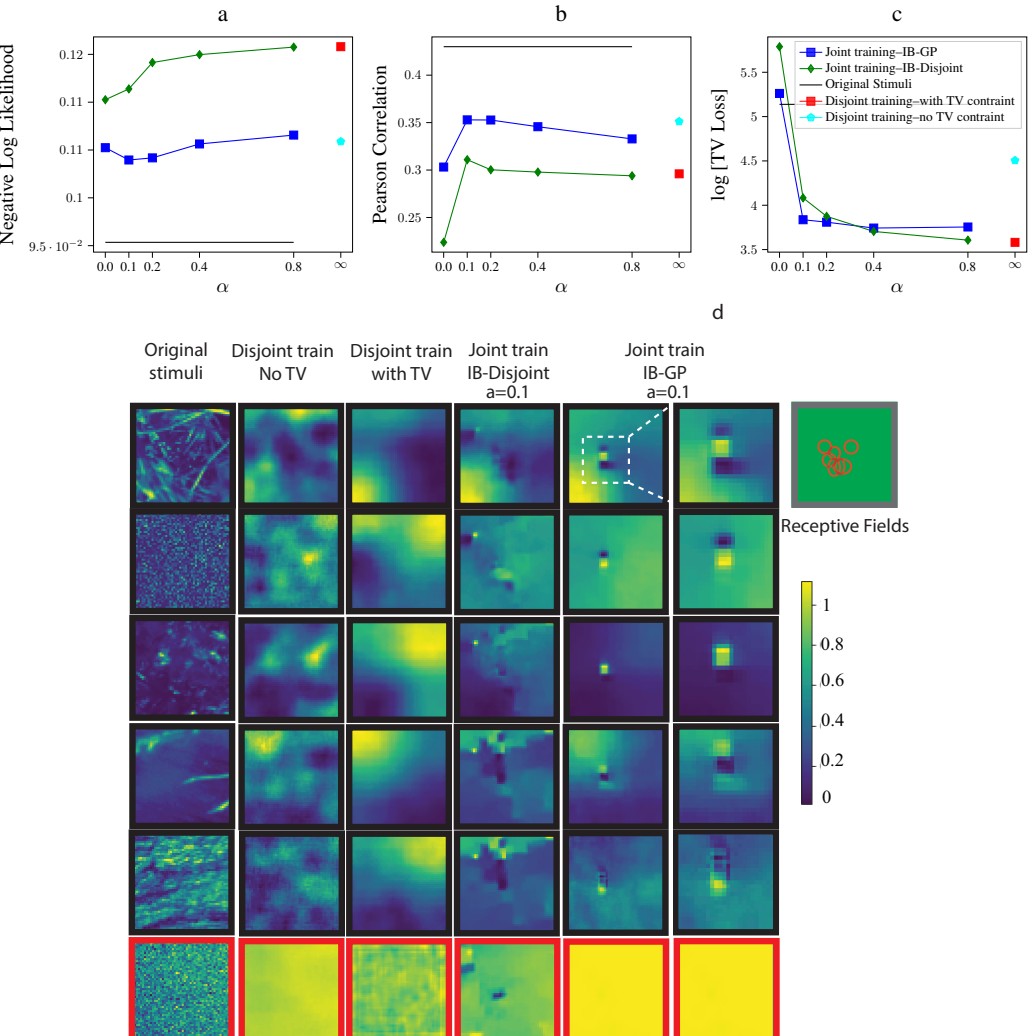

Figure 4: Summary of the closed-loop image synthesis experiment. The image synthesizer is trained either with the forward model (joint training) or without the forward model in the loop (disjoint training). We used both the IB-GP and IB-Disjoint as the forward model in the former case. In the latter case, the synthesizer is trained only to reconstruct the high-dimensional stimuli without incorporating any information from RGCs. The disjoint training is either subject to a smoothness constraint on the reconstructed images or with no constraint. (a) Poisson negative log-likelihood and (b) Pearson correlation for the RGC neural activity predictions. (c) TV loss for the synthesized stimuli. Performances of the original stimuli in the true model are depicted as a flat black line. Joint training with IB-GP with $\alpha = 0.1$ as the forward model achieves the best performance among all cases. (d) Examples of the synthesized images in all scenarios. IB-GP synthesized images contain particular patterns that are within the RFs of cells. The RFs of the neurons are depicted at scale.

without neural information results in non-smoothed images (column 2) or smoothed images (column 3) but poor RGC response accuracy (rightmost results in Fig. 4a-c). On the contrary, incorporating RGC neural information obtains synthesized stimuli with excellent smoothness and significant RGC response accuracy (Fig. 4a-c and columns 5,6 in F4d). These results demonstrate the robustness of the proposed algorithm in finding synthesized images that provide accurate responses. Strikingly, when IB-GP is used as the forward model in the closed-loop experiment, complex curvature shapes appear in the synthesized images. Expectedly, these shapes appear where the neurons' receptive field is located. The receptive fields (RFs) of the neurons, obtained by Spike-triggered Average (STA) analysis [1], are also shown in the figure. Finally, we continue the optimization of the image synthesizer for the IB-GP model with hyperparameter $\alpha = 0.1$ for another two rounds. The Pearson

correlation of the synthesized stimuli in all three rounds reads as: 0.353, 0.366, and 0.361. Details of the iterative optimization and evolution of the synthesized images in each phase are explained in Appendix 6. The last row in the Figure 4d shows a failure mode of the image synthesizer algorithm. We tested the image synthesizer (trained on the Natural dataset) on the white noise data. As seen from the figure, IB-Disjoint provides a superior synthesized image than other methods. This is in line with our observations that the IB-GP model fails to fit neural data elicited by the white noise. More results are available in Appendix 8.

## 7 Discussion

This work presents the IB-GP model, a latent space variational model that learns a low-dimensional representation of the Natural stimuli and RGCs' responses. We observed that the IB formulation of the neural activity fitting allows learning of a low-dimensional representation of the stimuli and RGC responses which is more predictive than the feedforward models. We demonstrated that incorporating a temporal prior on the latent space of the IB model factorizes the sources of neural variability into fewer latents. The Latent analysis of the model revealed the superior performance of the IB-GP to IB-Disjoint with no temporal prior on the latents. We observed that the dynamics of the high-dimensional stimuli and RGCs' responses were very well represented in the embeddings of the IB-GP model.

The model was then used in a closed-loop experiment to synthesize stimuli that elicit neuronal responses as those elicited by the original complex stimuli. Previously published work in the literature also looked into closed-loop image synthesis, mainly to find optimal stimuli that maximally activate neurons in the retina [24], or more recently in V1 and V4 regions in the mouse [34, 4]. The goal is to obtain spatial features that do not necessarily resemble the original stimuli but still elicit the same responses. This allows us to correctly use the resources available, for example, by placing the stimulation energy at suitable locations and with the correct spatiotemporal features. We observed various types of optimal stimuli for the Natural dataset, such as Gabor-like filters, on- and off-center Gaussians, or a mixture of both.

Given the promising results demonstrating the usefulness of the model, the IB-GP model has some limitations too. First, it is unclear what type of temporal prior is sufficient for the best factorization of latents. Although the Cauchy kernel has proved efficient in modeling multi-scale dynamics [13], it might be that other temporal kernels, a combination of several kernels, or even spatio -temporal kernels [9] could improve disentangling of the latents. Second, we observed the failure mode of the IB-GP model in the fitting of the white noise stimuli due to the lack of temporal correlation. Third, computing the covariances of the IB-GP's prior is computationally intensive, especially for the long temporal datasets. This might also be a possible negative societal impact of our work, resulting in a large carbon footprint.

## Acknowledgments and Disclosure of Funding

The authors would like to thank all the anonymous reviewers for their constructive feedback helping us to improve the manuscript, and especially for bringing the initial overfitting in Figure 2 to our attention, which allowed us to resolve it and improve the results. Authors declare no competing financial interests or funding.

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
