# Supplementary information for:
# Natural image synthesis for the retina with variational information bottleneck representation

**Babak Rahmani**
rahmani.b91@gmail.com

**Demetri Psaltis**
EPFL
demetri.psaltis@epfl.ch

**Christophe Moser**
EPFL
christophe.moser@epfl.ch

## 1 Information Bottleneck Gaussian Process formalism

To obtain a bound on the Information Bottleneck Gaussian Process (IB-GP) objective, we use the Markov chain constraint $\mathbf{Y} \leftrightarrow \mathbf{X} \leftrightarrow \mathbf{Z}$ and the factorized joint distribution [2]:

$$p(\mathbf{X}, \mathbf{Y}, \mathbf{Z}) = p(\mathbf{Y}|\mathbf{X}, \mathbf{Z})p(\mathbf{Z}|\mathbf{X})p(\mathbf{X}) = p(\mathbf{Y}|\mathbf{X})p(\mathbf{Z}|\mathbf{X})p(\mathbf{X}) \tag{1}$$

to expand the mutual information terms in $\mathcal{L}_{IB} = \max\big[\,I(\mathbf{Z}, \mathbf{Y}) - \beta I(\mathbf{Z}, \mathbf{X})\big]$. Henceforth, we use the stochastic encoder $p_\phi(\mathbf{Z}|\mathbf{X})$ parameterized by $\phi$ as an approximation for $p(\mathbf{Z}|\mathbf{X})$. Starting with $I(\mathbf{Z}, \mathbf{X})$, we have:

$$
\begin{aligned}
I(\mathbf{Z}, \mathbf{X}) &= \int d\mathbf{X}\, d\mathbf{Z}\, p(\mathbf{X}, \mathbf{Z})\, \log \frac{p(\mathbf{X}, \mathbf{Z})}{p(\mathbf{X})p(\mathbf{Z})} \\
&= \int d\mathbf{X}\, d\mathbf{Z}\, p(\mathbf{X}, \mathbf{Z}) \log p(\mathbf{Z}|\mathbf{X}) - \int d\mathbf{X}\, d\mathbf{Z}\, p(\mathbf{X}|\mathbf{Z})p(\mathbf{Z}) \log p(\mathbf{Z}) \\
&= \int d\mathbf{X}\, d\mathbf{Z}\, p(\mathbf{X}, \mathbf{Z}) \log p(\mathbf{Z}|\mathbf{X}) - \int d\mathbf{Z}\, p(\mathbf{Z}) \log p(\mathbf{Z})
\end{aligned}
\tag{2}
$$

where the second term on the RHS of Eq. 2 is the entropy $H(\mathbf{Z})$. In practice computation of $H(\mathbf{Z})$ might be intractable (even though $P(\mathbf{Z})$ is well defined). Therefore, a variational approximation $\rho(\mathbf{Z})$ is used in place of $p(\mathbf{Z})$ such that $\mathrm{KL}(p(\mathbf{Z}), \rho(\mathbf{Z}))$ is minimal. Therefore, with $\mathrm{KL}(p(\mathbf{Z}), \rho(\mathbf{Z})) \geq 0$, we have:

$$
\begin{aligned}
I(\mathbf{Z}, \mathbf{X}) &= \int d\mathbf{X}\, d\mathbf{Z}\, p(\mathbf{X}, \mathbf{Z}) \log p(\mathbf{Z}|\mathbf{X}) - \int d\mathbf{Z}\, p(\mathbf{Z}) \log p(\mathbf{Z}) \\
&\leq \int d\mathbf{X}\, d\mathbf{Z}\, p(\mathbf{X}, \mathbf{Z}) \log p(\mathbf{Z}|\mathbf{X}) - \int d\mathbf{Z}\, p(\mathbf{Z}) \log \rho(\mathbf{Z}) \\
&= \int d\mathbf{X}\, d\mathbf{Z}\, p(\mathbf{X}|\mathbf{Z})p(\mathbf{Z}) \log \frac{p(\mathbf{Z}|\mathbf{X})}{\rho(\mathbf{Z})} = \int d\mathbf{X}\, d\mathbf{Z}\, p(\mathbf{Z}|\mathbf{X})p(\mathbf{X}) \log \frac{p(\mathbf{Z}|\mathbf{X})}{\rho(\mathbf{Z})}\;.
\end{aligned}
\tag{3}
$$

Using the stochastic encoder $p_\phi(\mathbf{Z}|\mathbf{X})$, an upper bound on $I(\mathbf{Z}, \mathbf{X})$ reads as:

$$I(\mathbf{Z}, \mathbf{X}) \leq \int d\mathbf{X}\, d\mathbf{Z}\, p_\phi(\mathbf{Z}|\mathbf{X})p(\mathbf{X}) \log \frac{p_\phi(\mathbf{Z}|\mathbf{X})}{\rho(\mathbf{Z})}\;. \tag{4}$$

Moving on to the term $I(\mathbf{Z}, \mathbf{Y})$, we have:

36th Conference on Neural Information Processing Systems (NeurIPS 2022).

$$I(\mathbf{Z}, \mathbf{Y}) = \int d\mathbf{Y} \, d\mathbf{Z} \, p(\mathbf{Y}, \mathbf{Z}) \, \log \frac{p(\mathbf{Y}, \mathbf{Z})}{p(\mathbf{Y})p(\mathbf{Z})}$$

$$= \int d\mathbf{Y} \, d\mathbf{Z} \, p(\mathbf{Y}, \mathbf{Z}) \log p(\mathbf{Y}|\mathbf{Z}) - \int d\mathbf{Y} \, p(\mathbf{Y}) \log p(\mathbf{Y}) \tag{5}$$

where the second term on the RHS of Eq. 5 is the entropy $H(\mathbf{Y})$. In practice computation of $p(\mathbf{Y}, \mathbf{Z})$ and $p(\mathbf{Y}|\mathbf{Z})$ might be intractable (even though they are well defined). From Eq. 1, $p(\mathbf{Y}, \mathbf{Z})$ is written as $p(\mathbf{Y}, \mathbf{Z}) = \int d\mathbf{X} \, p(\mathbf{Y}|\mathbf{X})p_\phi(\mathbf{Z}|\mathbf{X})p(\mathbf{X})$. Additionally, a variational approximation $q_\theta(\mathbf{Y}|\mathbf{Z})$ is used in place of $p(\mathbf{Y}|\mathbf{Z})$ such that $\mathrm{KL}(q_\theta(\mathbf{Y}|\mathbf{Z}), p(\mathbf{Y}|\mathbf{Z})$ is minimal. Therefore, with $\mathrm{KL}(q_\theta(\mathbf{Y}|\mathbf{Z}), p(\mathbf{Y}|\mathbf{Z})) \geq 0$, we have:

$$I(\mathbf{Z}, \mathbf{Y}) = \int d\mathbf{Y} \, d\mathbf{Z} \, p(\mathbf{Y}, \mathbf{Z}) \log p(\mathbf{Y}|\mathbf{Z}) + H(\mathbf{Y})$$

$$\geq \int d\mathbf{Y} \, d\mathbf{Z} \, d\mathbf{X} \, p(\mathbf{Y}|\mathbf{X})p_\phi(\mathbf{Z}|\mathbf{X})p(\mathbf{X}) \log q_\theta(\mathbf{Y}|\mathbf{Z}) + H(\mathbf{Y}) \tag{6}$$

With the bounds on $I(\mathbf{Z},\mathbf{Y})$ and $I(\mathbf{Z}, \mathbf{X})$, the IB objective reads as:

$$\mathcal{L}_{IB} = \max\big[\, I(\mathbf{Z}, \mathbf{Y}) - \beta I(\mathbf{Z}, \mathbf{X})\big] \geq \max_{\theta,\phi}\big[\, I_{IB}\big]$$

where

$$I_{IB} = \int d\mathbf{Y} \, d\mathbf{Z} \, d\mathbf{X} \, p(\mathbf{Y}|\mathbf{X})p_\phi(\mathbf{Z}|\mathbf{X})p(\mathbf{X}) \log q_\theta(\mathbf{Y}|\mathbf{Z}) - \beta \int d\mathbf{X} \, d\mathbf{Z} \, p_\phi(\mathbf{Z}|\mathbf{X})p(\mathbf{X}) \log \frac{p_\phi(\mathbf{Z}|\mathbf{X})}{\rho(\mathbf{Z})} \tag{7}$$

As explained in the main text, we assume the joint distribution $p(\mathbf{X}, \mathbf{Y})$ is approximated by:

$$p(\mathbf{X}, \mathbf{Y}) = \frac{1}{T} \sum_{t=1}^{T} \delta(\mathbf{X} - x_{1:\tau(t)})\delta(\mathbf{Y} - y_t), \tag{8}$$

where $\tau(t)$ is to represent time dependence up to time $t$. The joint distribution $p(\mathbf{X}, \mathbf{Y})$ presented above allows the lower bound $I_{IB}$ to be approximated by:

$$I_{IB} \geq \frac{1}{T} \sum_{t=1}^{T} \left[ \int d\mathbf{Z} \, p_\phi(\mathbf{Z}|x_{1:\tau(t)}) \log q_\theta(y_t|\mathbf{Z}) - \beta \int d\mathbf{Z} \, p_\phi(\mathbf{Z}|x_{1:\tau(t)}) \log \frac{p_\phi(\mathbf{Z}|x_{1:\tau(t)})}{\rho(\mathbf{Z})} \right]$$

$$= \frac{1}{T} \sum_{t=1}^{T} \left[ \mathbb{E}_{p_\phi(\mathbf{Z}|x_{1:\tau(t)})}[\log q_\theta(y_t|\mathbf{Z})] - \beta D_{KL}[p_\phi(\mathbf{Z}|x_{1:\tau(t)})||\rho(\mathbf{Z})] \right] \tag{9}$$

Finally, we enforce the Gaussian Process (GP) prior to derive the IB lower bound:

$$I_{IB} \geq \frac{1}{T} \sum_{t=1}^{T} \left[ \mathbb{E}_{p_\phi(\mathbf{Z}|x_{1:\tau(t)})}[\log q_\theta(y_t|\mathbf{Z})] - \beta D_{KL}[p_\phi(\mathbf{Z}|x_{1:\tau(t)})||\mathcal{N}_\mathbf{Z}(\mathbf{0}, \mathbf{K})] \right]. \tag{10}$$

where $\mathbf{K} \in \mathbb{R}^{\mathcal{T} \times \mathcal{T}}$ is the GP's covariance.

## 2   Training specifications and networks architecture

All models in the paper are implemented in Tensorflow [1]. We used Adam optimizer [5] with a constant learning rate of $10^{-4}$ ($\beta_1 = 0.9, \beta_2 = 0.999$). Training of the models is carried out for a fixed number of epochs (see tables below). To avoid over-fitting of the models across different architectures and hyperparameters, we used Poisson negative log likelihood of the dataset (rather than the Pearson correlation) as the flag for early stopping when training each model individually. [1] At test time, weights and biases of the networks were initialized to those which provided the best loss on the validation set. The input images were scaled to have values between 0 and 1.

The hyperparameters of the models used for benchmarking of the data are summarized in Table 1. Architectures of the networks are presented in Tables 2-3. The batch-norm layer in the architecture of the networks is based on the vanilla batch-norm in [4]. We used two different architectures to implement the IB-GP (IB-Disjoint) model. In the first architecture, the dimension of the data in the temporal direction is kept unchanged. In the second architecture, the data dimension in the temporal dimension is first reduced by a factor of two in the encoder and then increased by the same factor in the decoder of the model. This reduces the temporal size of the latent variables to half of the original data's temporal size. We did not notice any noticeable change in the performances of the models for these two architectures. However, the compute time of the second architecture is less than the first one. All results presented in this work are obtained using the second architecture except for the results in IB-GP's and IB-Disjoint's latent analysis section.

The image synthesizer is implemented as an Auto-encoder network with a latent size of 15 (Table 4). To choose the latent size of the Auto-encoder, we applied a one-component Principal Component Analysis (PCA) to the high-dimensional stimuli and measured the Pearson correlation of the responses elicited by the PCA-reconstructed stimuli when fed to the true model in the closed-loop experiment (Disjoint scenario). Then we chose the latent size of the Auto-encoder such that responses elicited by the images that were reconstructed by the Auto-encoder in the Disjoint scenario (with a TV smoothness constraint) obtain a Pearson correlation around the same value as that of the PCA-reconstructed images (Pearson correlation $\sim 0.291$). This way, we ensure we obtain a model agnostic compression of the stimuli by being objective to the choice of the image synthesizer architecture.

|  | Feedforward CNN [6] | IB-Disjoint and IB-GP |
|---|---|---|
| Number of training epochs | 200 (Natural) \| 300 (Brownian) | 200 (Natural) \| 300 (Brownian) |
| Number of samples in each batch | 1000 (Natural) \| 600 (Brownian) | 1000 (Natural) \| 600 (Brownian) |

Table 1: Training hyperparameters used for obtaining the results in the Figure 2 of the main paper.

---

[1]We note that the train set's RGCs' responses of the Natural dataset are count data (integer numbers), whereas the test set's RGC responses are averaged responses of repeated numbers. This results in the Pearson correlation of the train set always being smaller than the Pearson correlation of the test set. Therefore, we observed that the Poisson negative log likelihood is a better metric for early stopping and avoiding over-fitting.

| Encoder | Decoder |
|---|---|
| Input $x \times x \times T$ seq. of imgs | F.C. output $16 \times 12 \times 12$ Relu |
| Batch-norm | Batch-norm |
| $3 \times 3$ conv. 64 s. 1 same Relu | $40 \times 1$ 1D-conv. 16 s. 1 same Relu |
| $2 \times 2$ maxpool | $2 \times 1$ 1D Upsampling |
| Batch-norm | Batch-norm |
| $3 \times 3$ conv. 32 s. 1 same Relu | $3 \times 3$ conv. 32 s. 1 same Relu |
| $2 \times 2$ maxpool | Batch-norm |
| Batch-norm | $2 \times 2$ Upsampling |
| $3 \times 3$ conv. 16 s. 1 same Relu | 4 sided zero pad. |
| Batch-norm | $3 \times 3$ conv. 64 s. 1 same Relu |
| $40 \times 1$ 1D-conv. 16 s. 1 same Relu | Batch-norm |
| $2 \times 1$ 1D maxpool | $2 \times 2$ Upsampling |
| Batch-norm | $3 \times 3$ conv. 1 s. 1 same Relu |
| F.C. output $2/3\times$ Latent dim. No activ. | Batch-norm |
| | $21 \times 21$ conv. 4 s. 1 no pad. no activ. |
| | $40 \times 1$ 1D-conv. 4 s. 1 same Relu |
| | $15 \times 15$ conv. 4 s. 1 no pad. Relu |
| | F.C. output $y$ Softplus activ. |

Table 2: IB-Disjoint/GP architecture. The IB-Disjoint/GP encoder output sizes are $2/3\times$latent dim., respectively. FC: Fully connected, same: same padding, conv: 2D convolution, 1D-conv: 1D convolution, maxpool: 2D max pooling layer, 1D maxpool: 1D max pooling layer, Softplus: softplus activation function, Upsampling: 2D up sampling layer, 1D up sampling: 1D up sampling layer.

| Feedforward CNN [6] |
|---|
| Input $x \times x \times T$ seq. of imgs |
| $21 \times 21$ conv. 4 s. 1 no pad. no activ. |
| Gaussian noise zero mean 0.1 standard variance. |
| $40 \times 1$ 1D-conv. 4 s. 1 same Relu |
| $15 \times 15$ conv. 4 s. 1 no pad. Relu |
| Gaussian noise zero mean 0.1 standard variance. |
| F.C. output $y$ Softplus activ. |

Table 3: Feedforward CNN architecture.

| Image synthesizer |
|---|
| Input $x \times x \times T$ seq. of imgs |
| $3 \times 3$ conv. 64 s. 1 same Relu |
| $2 \times 2$ maxpool |
| $3 \times 3$ conv. 32 s. 1 same Relu |
| $2 \times 2$ maxpool |
| $3 \times 3$ conv. 16 s. 1 same Relu |
| F.C. output $2\times$ 15. No activ. |
| F.C. output $16 \times 12 \times 12$ Relu |
| $3 \times 3$ conv. 32 s. 1 same Relu |
| $2 \times 2$ Upsampling |
| 4 sided zero pad. |
| $3 \times 3$ conv. 64 s. 1 same Relu |
| $2 \times 2$ Upsampling |
| $3 \times 3$ conv. 1 s. 1 same Sigmoid |

Table 4: Image synthesizer architecture.

# 3 Examples of the input stimuli in the datasets

Examples of the input stimuli in the Natural and Brownian datasets are depicted in Supplementary Figure S1.

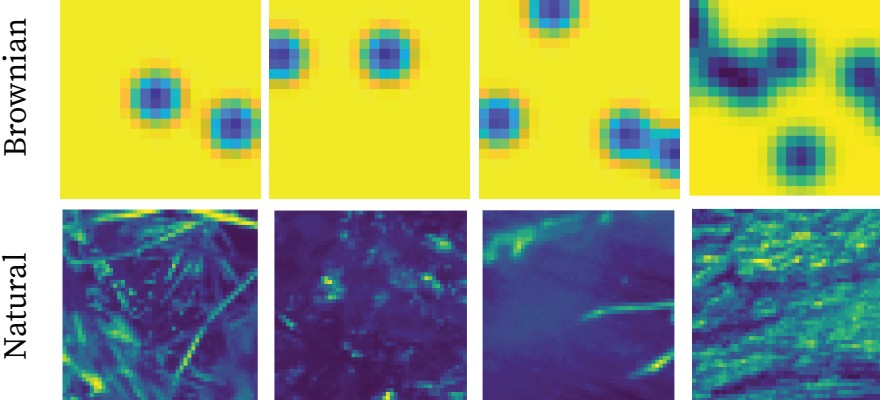

Figure S1: Examples of the stimuli in the Brownian (top row) and Natural (bottom row) datasets. We note the similarity among stimuli that justifies using a prior temporal correlation in the IB-GP model.

# 4 IB-GP and IB-Disjoint neural activity predictive performance in the low-value prior constraint regime

Figure S2 shows the predictive performances of the IB-GP and IB-Disjoint models versus the number of latents for both the Natural and Brownian datasets when the models are trained in the low-value prior constraint regime (small values of $\beta$). In this regime, IB-Disjoint obtains a superior performance than the IB-GP model. Despite this superior performance, latent space of the IB-Disjoint is more spread than the IB-GP.

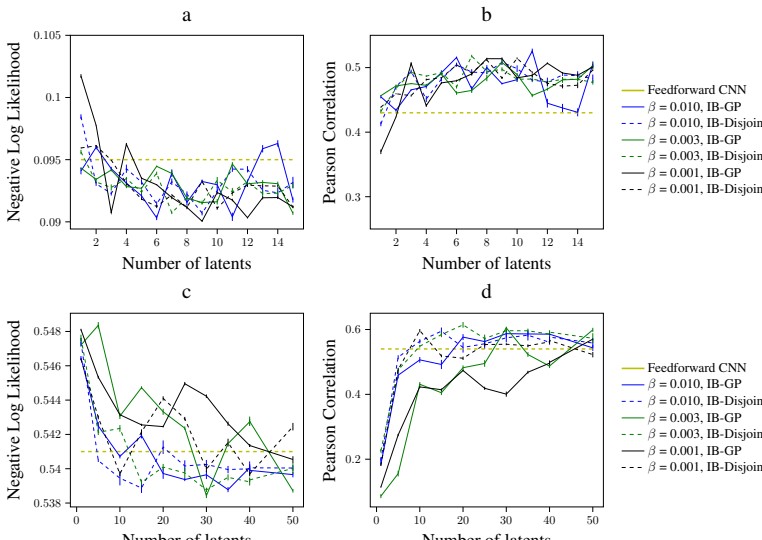

Figure S2: Overview of the RGCs' neural activity prediction accuracy for the IB-GP, IB-Disjoint, and Feedforward CNN trained on the Natural and Brownian dataset. (a) Poisson negative log-likelihood of the predicted neural responses and the (b) Pearson correlation of the predicted and true spike responses as a function of the number of latents for different values of the prior pressure constraint $\beta$ for the Natural dataset. (c, d) the same as (a,b) but for the Brownian dataset. In each plot, the flat line shows the performance of the Feedforward CNN.

# 5 Brownian dataset

## 5.1 IB-GP's predictive performance for a varying number of neurons

We trained the IB-GP model with a varying number of neurons in the Brownian dataset to show that the results are not dependent on the number of neurons. The neurons were selected based on a stability criterion in which a cell is deemed stable if the Pearson correlation between its average response to the same stimulus over different blocks of trials exceeds a threshold [6]. Specifically, we used RGCs' responses of 9, 15, 21, 27, 36 and 51 neurons (corresponding to the thresholds of 0.70, 0.65, 0.55, 0.50 and 0.3) to train the IB-GP model. While the 9-neuron case is covered in the main paper, Figure S3 plots the Pearson correlation of the predicted and the ground truth responses. As can be seen in the figure, the IB-GP model can fit neural responses of the RGCs for different values of the neurons. We also observe that the model's predictive performance decreases with increasing the number of neurons. The latter is related to the fixed latent size of the model.

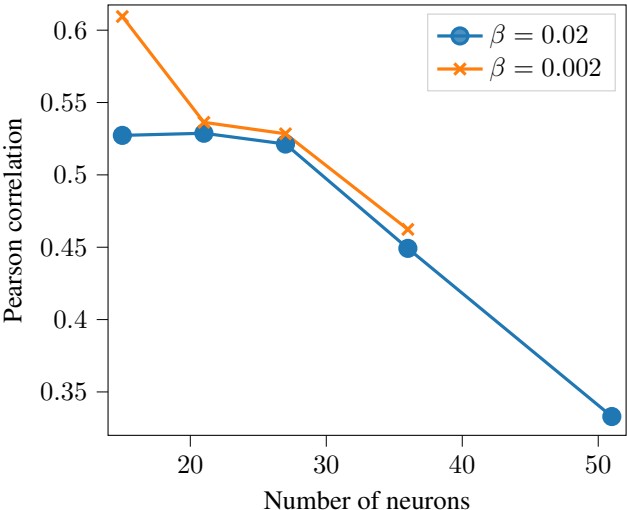

Figure S3: Pearson correlation of the predicted and true spike responses as a function of the number of neurons in the Brownian dataset for two values of the prior pressure constraint $\beta$. The IB-GP is trained with a fixed latent dimension size of 50.

## 5.2 Comparison of the IB-GP and IB-Disjoint learned dynamics

We analyzed the learned dynamics of the IB-GP and IB-Disjoint trained with 50 latents and $\beta = 0.02$ on the *Brownian dataset*. The variability in the accuracy of the models for the traversed latent factors is depicted as blue shades in Figure S4(a,d). The KL divergence for each latent is also plotted. As observed in the figure, the latent information in the IB-GP model is less distributed over the latent space than in the IB-Disjoint (blue shades in a and d). Moreover, Figure S4(b,e) visualizes the inferred dynamics of the most informative latent in each model (latent 6 in IB-GP and 34 in IB-Disjoint) after reducing the dimension to 2 with T-SNE. The dynamics of the most informative latent in the IB-GP have three clusters. These three clusters might be related to the three types of stimuli in the Brownian dataset[2]. The Auto-correlation of the most informative latent in each model, as well as the Auto-correlation of the true activities and predicted responses (averaged over the neurons), are depicted in Figure S4(c,f). Finally, the predicted neural activities of the models for an example neuron are plotted in Figure S4(g,h).

---

[2]The Brownian dataset consists of four types of stimuli with 1, 2, 4, and 10 discs in the visual field. We did not use the one-disc stimuli as they happened to contain instances of images with no discs in the visual field due to the Brownian movement[3]

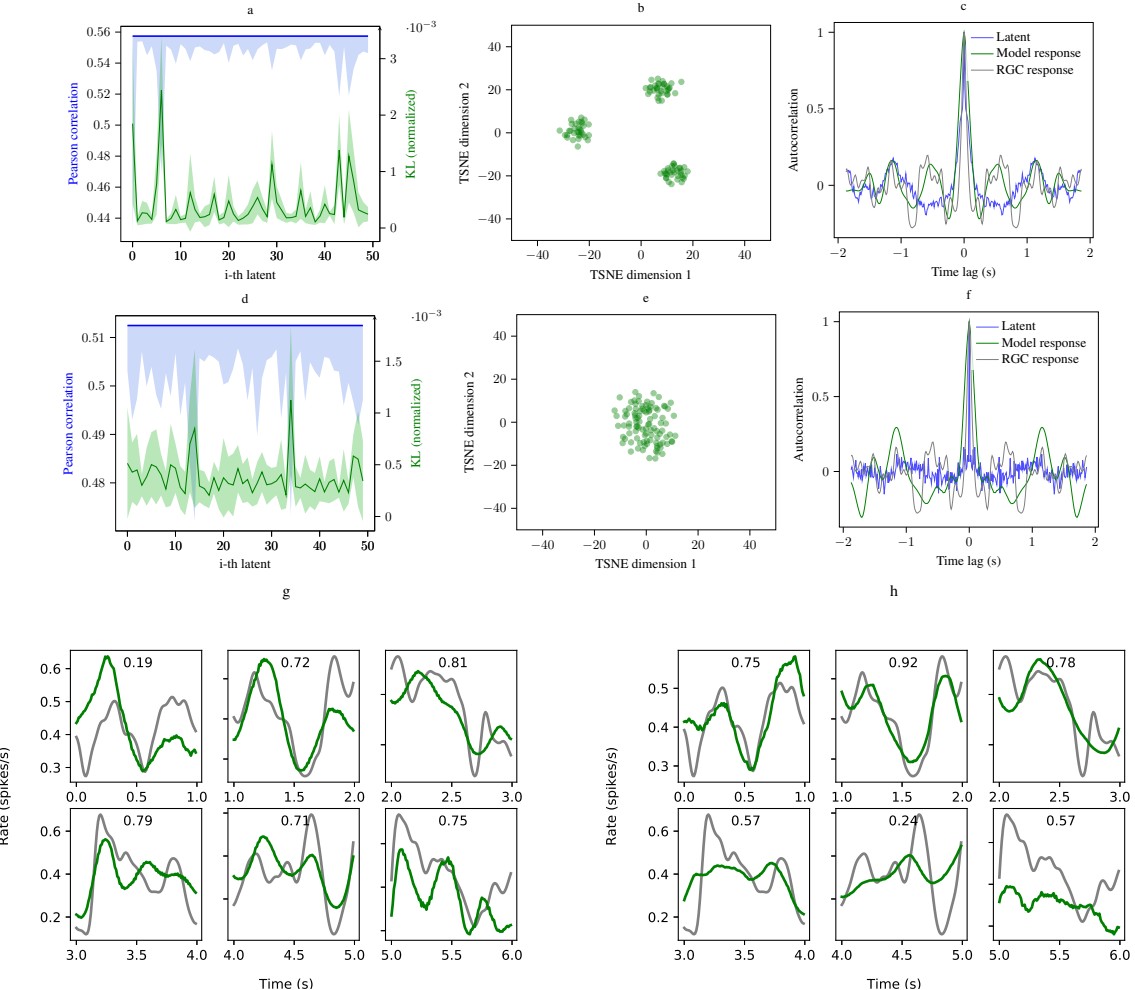

Figure S4: Analysis of the IB-GP (first row) and IB-Disjoint's (second row) learned dynamics as well as neural predictive performance (IB-GP: third row left, IB-Disjoint: third row right) for the Brownian dataset. (a,d) The average prediction accuracy of the IB-GP (a) and IB-Disjoint (d), measured by the Pearson correlation, is shown as a flat line. The variability in the accuracy of the model for traversed latent factors is depicted as blue shades. Averaged normalized KL divergence of each latent is also plotted. (b,e) The inferred dynamics of the most informative latent variable from (a,d) are visualized after reducing the dimensions to 2 with T-SNE. (c,f) Auto-Correlation of the most informative latent is compared against those of the true temporal activity and the model's predicted responses (averaged over neurons). (g,h) Neural activity prediction of the models. Numbers show the correlation of the predicted responses with the true activities.

## 5.3 Single neuron Auto-correlations for the IB-GP model trained on the Brownian dataset

The Auto-correlation dynamics of the single neurons are depicted in Figure S5.

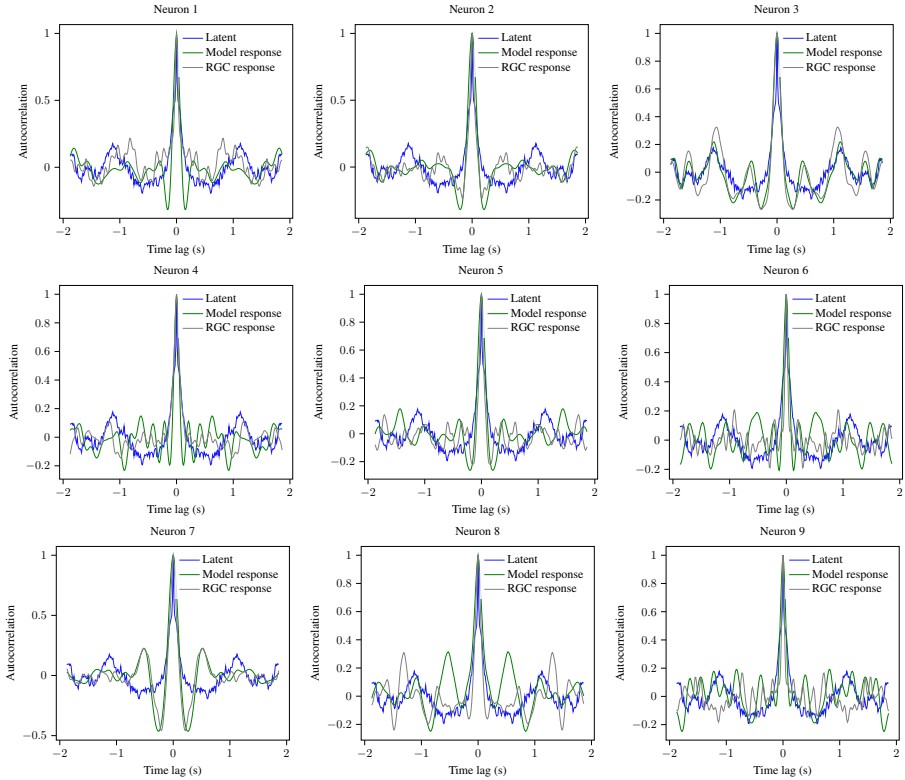

Figure S5: Autocorrelation of the most informative latent in the IB-GP model is compared against those of the true temporal activity and the model's predicted response for each neuron in the Brownian dataset with 9 RGCs responses.

## 5.4 Closed loop image synthesis

Figure S6 plots the synthesized stimuli of the closed-loop image synthesis algorithm and the IB-GP model that is trained on the Brownian dataset. The receptive fields (RF) of the neurons are also plotted. The minimal information stimuli obtain a Pearson correlation of 0.51 and a negative log-likelihood of 0.543. The original stimuli obtains a Pearson correlation of 0.55 and a negative log-likelihood of 0.541 It can be seen that the algorithm finds the optimal stimulus that is maximally activating the neurons within their RF.

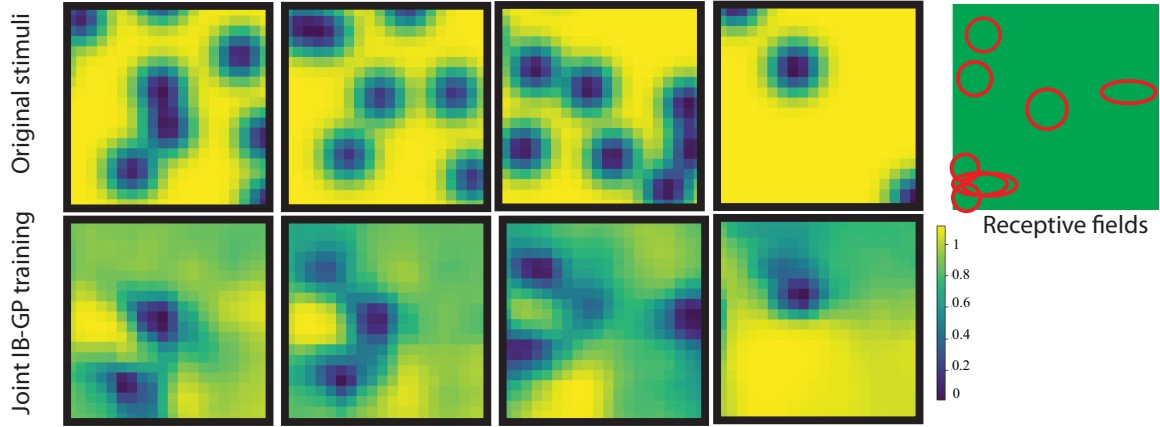

Figure S6: Examples of the synthesized stimuli using the IB-GP model with $\beta = 0.02$. Original stimuli are also shown. The data is from the Brownian dataset with 9 neurons. The RFs of the neurons in the dataset are depicted at scale.

# 6 Natural dataset

## 6.1 IB-GP model trained on the Natural dataset

Below, we further analyze the latent dynamics of the IB-GP model discussed in section 6.1 (15 latents, $\beta = 0.05$).

**Visualization of the learned covariance matrices and the traversal analysis of the latents** In Figure S7, we visualize the learned covariance matrices of the first two most informative latents. The least informative latent of the IB-GP model and the Cauchy kernel prior are also depicted. We note the diagonally banded structure of the covariance matrices. For the most informative latent, the structure along the diagonal of the covariance matrix shows local dependency among the stimuli and elicited neural responses. We observe regions along the diagonal that are similar to the prior. These regions appear because of the stimuli with no temporal dependency (white noise stimuli interleaved in the test dataset) or stimuli that do not contain any information within the receptive fields of the neurons. In the latter case, the RGCs' responses are mainly stochastic. For the second most informative latent, the structure of the covariance matrix mostly resembles that of the prior except for regions with minor variations. This illustrates that the first informative latent picks up most of the information required for explaining the neural codes of the RGCs. In contrast, the second most informative latent attends to the higher-order information. The traversal analysis of the latents corroborates these findings. The traversal plots of the IB-GP model for the first and the second most informative latents are plotted in Figure S8. The true spiking activities of an example neuron are plotted in gray. The predicted responses of the model with values of all latents set to their inferred ones except for one latent variable whose value is being traversed are plotted in green. As shown in the figure, the most informative latent learns to represent the timings of the RGCs as the stimuli appear. In contrast, the second informative latent picks up the information about the exact firing rates of the RGCs.

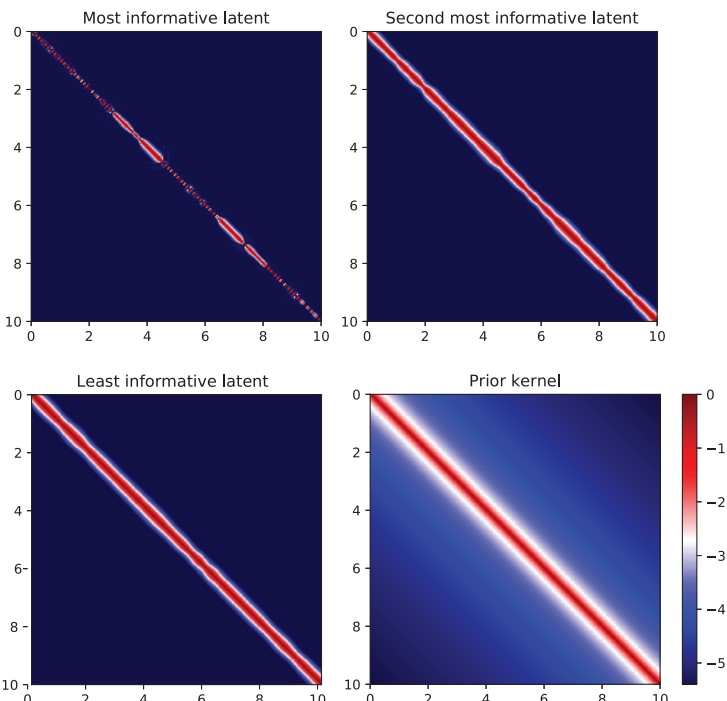

Figure S7: Learned covariance matrices of the first two most informative and the least informative latents in the IB-GP model. The axes are time values in seconds. We note that how the local time dependencies are visible along the main axis of the most informative latent covariance matrix. The Cauchy kernel is plotted for comparison.

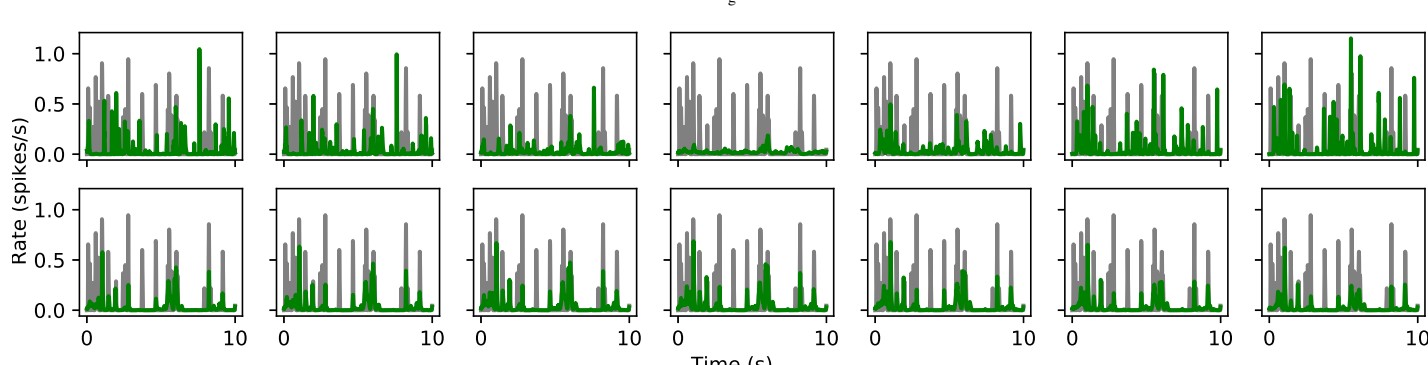

Figure S8: Traversal plots of the IB-GP model for the first (top row) and second (bottom row) most informative latents for an example neuron. All plots contain the true spiking responses of an example neuron in gray. The predicted responses of the model with values of all latents set to their inferred ones except for one latent variable whose value is being traversed are plotted in green.

## 6.2 Single neuron Auto-correlations for the IB-GP model trained on the Natural dataset

Auto-correlation dynamics of the true activities, the predicted responses, and the most informative latent of the IB-GP model for all neurons are depicted in Figure S9.

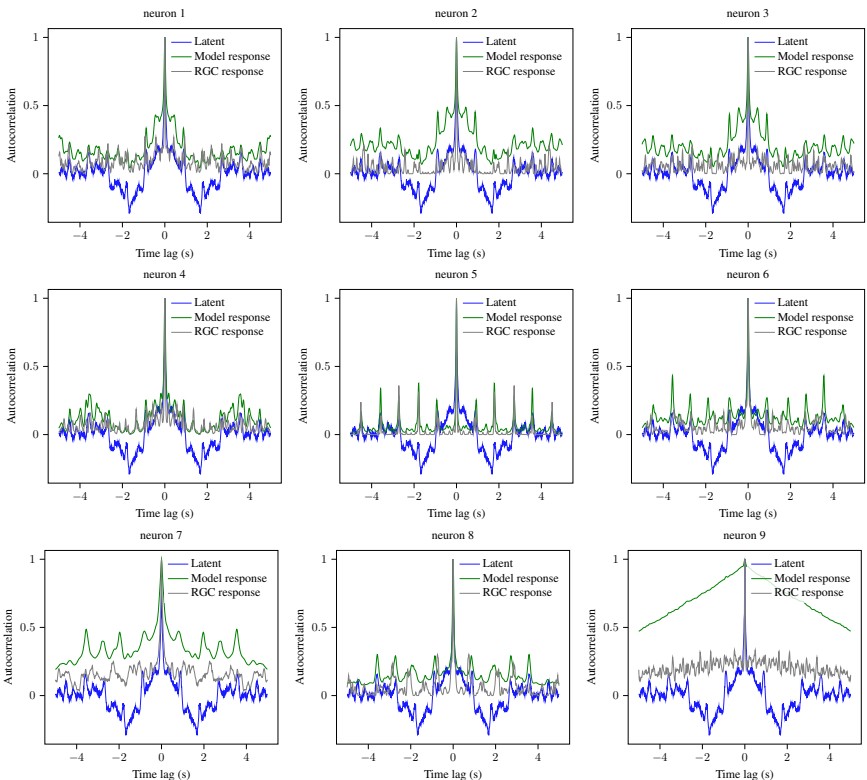

Figure S9: Autocorrelation of the most informative latent in the IB-GP model is compared against those of the true temporal activity and the model's predicted response for each neuron in the Natural dataset with 9 RGCs responses.

## 6.3 Closed loop image synthesis

Additional examples of the synthesized images from the Natural dataset are plotted in Figure S10. The images are zoomed around the receptive fields of the neurons. Additionally, we continued the optimization of the image synthesizer for the Natural dataset (with IB-GP as the forward model) for two additional rounds. The average Pearson correlation of the synthesized stimuli in all three rounds reads as: 0.353, 0.366, and 0.361. Figure S11 depicts some examples of the synthesized images in each iteration. From iteration one to two, some synthesized images undergo a trivial change (first row) or an averaged luminance change in the image background (second and third rows). We observed a decrease in the correlations of the neural responses elicited by the synthesized images in iteration three. The stimuli which were a failure mode of the IB-GP are of particular interest. We note that the image synthesizer puts special attention to the region of the neurons' receptive fields. However, it fails to synthesize the images with responses correlated with the original activities.

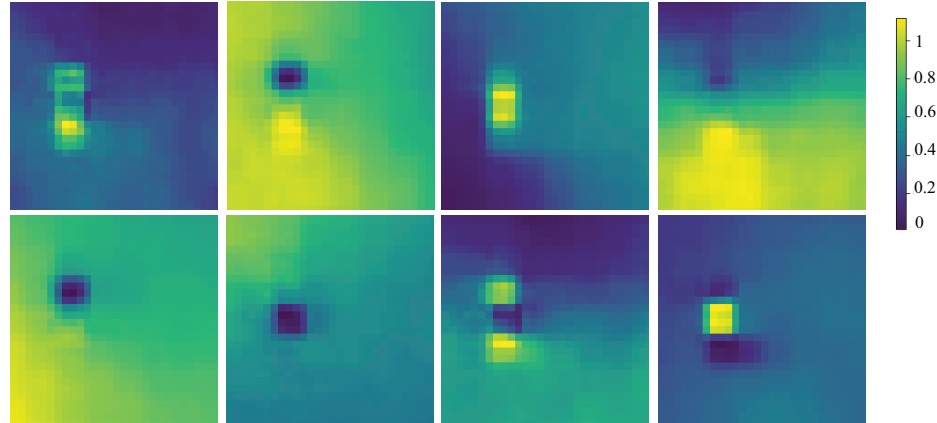

Figure S10: Examples of the synthesized stimuli from the Natural dataset for a one-round closed-loop iterative optimization. The stimuli are cropped around the RFs of the cells.

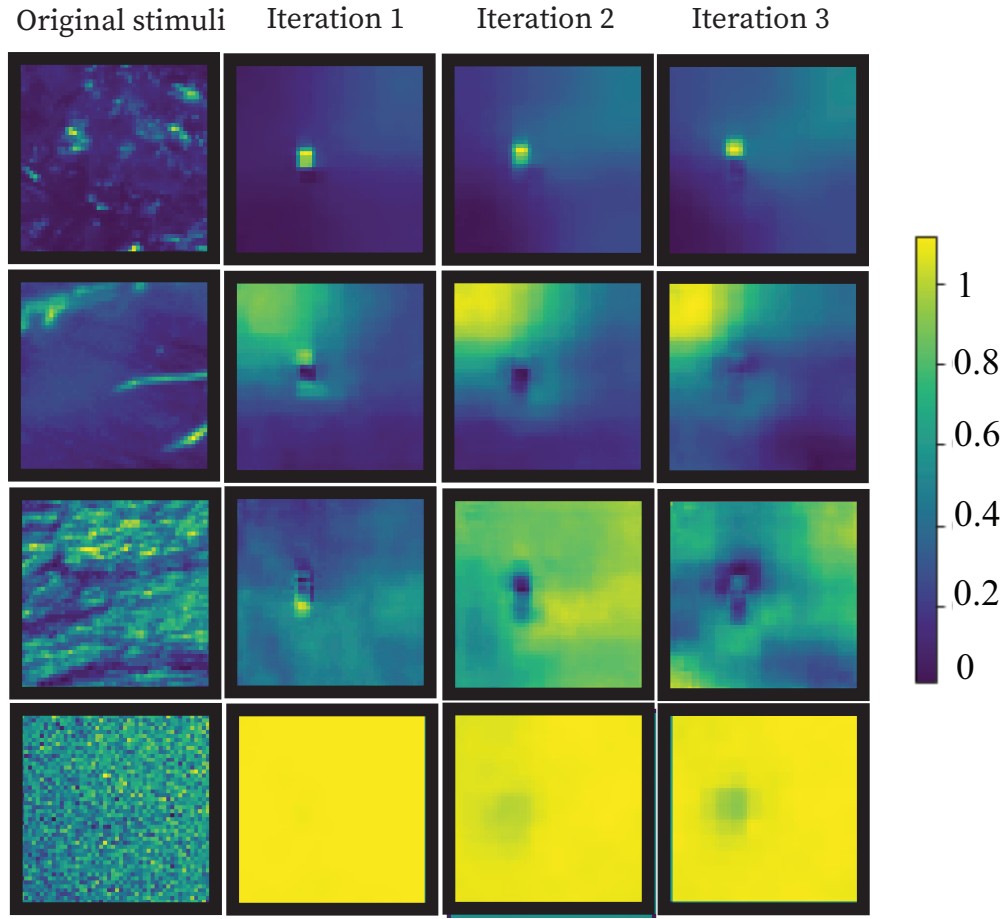

Figure S11: Examples of the synthesized stimuli from the Natural dataset for a three-round closed-loop iterative optimization.

## 7 Regularized and non-regularized image synthesis

Figure S12 plots examples of the synthesized images with and without the smoothness constraint.

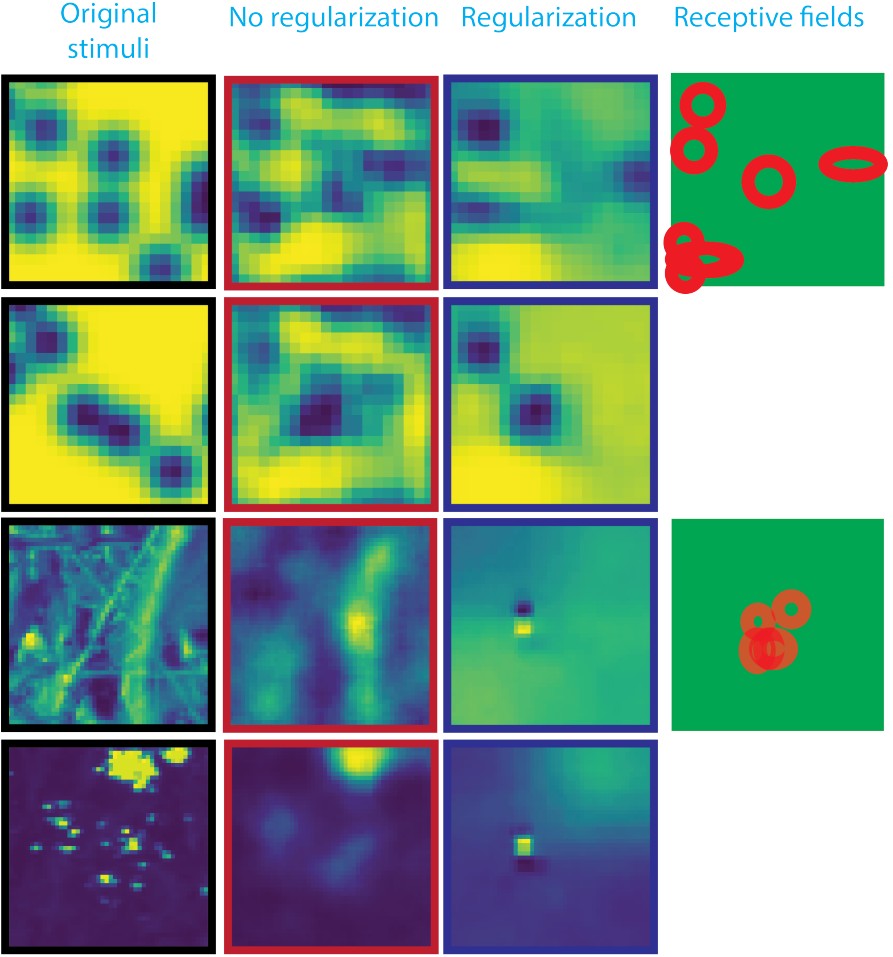

Figure S12: Regularized versus non-regularized image synthesis for the Brownian (rows 1 and 2) and Natural (rows 3 and 4) dataset.

## 8 Fitting of the neural activity to the white noise stimuli

We trained the IB-GP and IB-Disjoint models on the white noise stimuli dataset [6]. We observed that the IB-GP model failed to fit the neural responses. Although IB-Disjoint obtained better predictive performance than IB-GP, it failed to reach the performance of the feedforward CNN.

## 9 Software and Data

All models are implemented in Tensorflow.v.2.3. and run on Nvidia RTX 3090 GPU. Training of the IB-GP model takes about 1 hour of compute on the GPU.

Code for training the IB-GP and IB-Disjoint models are available at: `https://github.com/Babak70/Natural_image_synthesis`.