# OpenReview forum: "Natural image synthesis for the retina with variational information bottleneck representation"
_NeurIPS.cc/2022/Conference — NeurIPS 2022 Accept_

### Official Review · Reviewer_6crN · 2022-07-06

**Rating:** 6
**Confidence:** 4
**Soundness:** 3 good
**Presentation:** 3 good
**Contribution:** 3 good

**Summary:**

In this manuscript the authors develop a new variant of the variational bottleneck architecture with temporal dependencies in the latent space and apply it to the prediction of retinal ganglion cell responses. The temporal dependencies are introduced by using a Gaussian process prior, i.e. by adding a prior covariance across time to the latent space variables. Their architecture is quite successful at predicting retinal responses beating a previous network without a bottleneck. Finally, the authors optimize images to be smooth and produce the same latent representation as original input stimuli, which results in much simpler stimuli that are expected to produce similar responses in the neurons.

**Questions:**

1) Could you please clarify why the image generation is helpful?
2) Is there any chance to use a larger dataset for this?

**Limitations:**

---

**Strengths And Weaknesses:**

The introduction of temporal dependencies into the latent is an interesting extension of this architecture, the implementation seems sensible and the predictive performance of the approach is good. However, the authors predict only a very small number of neurons and I am a bit lost to why the image generation process is helpful. Thus, this paper seems like a sensible step forward, but not like the solution to the visual prosthetics problem as it is framed by the authors.

Concrete points:
1) The authors fit only 9 neurons in the main paper and up to 36 neurons in the supplement. This is extremely small. Large retinal recording datasets record from hundreds of ganglion cells electrophysiologically (https://doi.org/10.1162/neco_a_01395 for example use more than a 1000 for decoding) or using calcium imaging (https://www.nature.com/articles/nature16468 for example). Especially when stimuli are generated that presumably aim to activate all or at least many retinal ganglion cells in an area, this seems like a substantial drawback.
2) Similarly, the stimuli used here are in generated from only 10 different images. This restriction may contribute to the success of the small networks here and casts a little doubt on whether the described techniques would generalize to new images.
3) The authors generate smooth images that are meant to generate similar responses as the original images. This strikes me as odd, because smoothness appears largely unrelated to the constraints stimulation by a prosthetic device would face. Wouldn’t it make more sense to build in such constraints rather than generating images?


Small points:
- Line 116: The bi-diagonal precision matrices here are simple, but not low rank.
- Equation (7) should really be a lower bound not an approximation.
- Quite a few of the figures have extremely small annotations, especially when printed. Please, invest some time to make them more readable, remove unnecessary whitespace and boxes, etc.

---

> ### Author Response · Authors · 2022-08-01
> **Response to the reviewer 6crN**
>
> # General
>
> We thank reviewer 6crN for the detailed and thoughtful review. We appreciate that the reviewer finds our model novel and a step forward for the future of prosthetic within neuroscience community. It seems that the main concerns of the reviewer are the applicability of the IB-GP to larger dataset and the intuition behind using the closed-loop image synthesis. We hope we can adequately address these concerns in the following responses. Before getting to the specific points, let us emphasize that one of the main contributions of this work is presenting a latent space representation learning approach for modeling of the RGCs responses that based on the principled Information Bottleneck framework that fits naturally to the problem of early visual modeling and hence we believe our work could be of interest for the neuroscience community.
>
> # Major points/Questions
>
> >**1- The authors fit only 9 neurons ...**
>
> Please see the response to Q2 here.
>
> >2- **Similarly, the stimuli used here are generated from only 10 different images...**
>
> The concern of the reviewer about generalization of the proposed model to new images appears to be based on the misunderstanding that the dataset containing only 10 category of images. However, **this is not the case**. The Natural dataset contains 395000 input images. Each 100 images are one unique image which are jittered in time. Therefore, there are 3950 unique images in the set. These images are from the *Natural images from the birthplace of the human* (Gasper Tkacik et al. 2011) and are completely different and not categorical. The test dataset contains 50 **new** unique images (50x100). One sample of test set has 10 unique images (10x100). The large number of unique images in the dataset and the small footprint of the network (see Appendix 3 in the supplementary) confirms that the network is not memorizing the dataset. We would be happy to clarify this point in the manuscript.
>
> >**3- The authors generate smooth images...?**
>
> We did not set smoothness as the main constraint for the image synthesizer algorithm. We observed that with no smoothness constraint, the image synthesizer converged to solutions with spurious patterns outside the receptive fields of the neurons. Therefore, we found it useful to add such term in the objective function.
>
> # Minor points
>
> We corrected the points in the manuscript.
>
> # Questions
>
> >1. **Could you please clarify why the image generation is helpful**
>
> Previously published work in the literature have looked into closed-loop image synthesis, particularly to find optimal stimuli that maximally activate neurons in the Retina (Paninski, Progress in brain research, 2007), or more recently in V1 and V4 regions in the mouse (Walker, Nature neuroscience, 2019)(Bashivan, science, 2019). The goal is to obtain spatial features that not necessarily resemble the original stimuli but still elicit the same responses. This allows to correctly use the resources available, for example by placing the stimulation energy at the right locations and with the correct spatiotemporally features. Therefore, using techniques that focus only on obtaining the RF locations of neurons such as STA, etc. is perhaps not the optimal strategy. The IB-GP synthesized images obtained by our model is doing the same task. For the Natural dataset, we observed different types of patterns: locally high frequency features, Gabor-like filters Gaussian-like patterns, etc. Our findings are consistent with the previous findings in the neuroscience literature that these types of patterns are those that maximally drive early visual activity. We would be very happy to add this discussion to the updated manuscript.

---

> > ### Author Response · Authors · 2022-08-01
> > **continued**
> >
> > >**2- Is there any chance to use a larger dataset for this?**
> >
> > This is a good point. We ran our experiment on the only publicly available dataset we could find or dataset that was shared with us. Given this, we validated our model on a varying number of neurons. Therein, we consistently observed the model is able to learn the RGCs' neural dynamics in each case. We believe this demonstrates the the potential of the IB-GP model and the image-synthesis procedure to be applied to larger dataset in the future work. We agree that the ultimate solution to the prosthetics problem requires activating larger population of neurons. However, the aim of this work is to first provide a theoretical and principled approach for modeling of the early visual system using a framework which we believe is better suited for modeling the neuronal activity in the retina or even downstream in the cortical region than the feedforward CNN. We would be happy to revise the manuscript to clarify any confusion about the contributions of our work with respect to the prosthetics. Given that, at the suggestion of the reviewer, we ran additional experiments on the Brownian dataset with more neuronal responses. To do so, we lowered the criteria (outlined in Appendix 6 in the supplementary) by which the stable neurons were selected to a value of 0.3. This value is the minimum standard in the literature (McIntosh et al. 2016). This lead to the number of neurons going up to the values 51. The performance of the IB-GP in this case is reported in the Table below.
> >
> > **Number of neurons**         **Pearson Correlation**           **Retinal Reliability**
> > ***
> >                51                             0.333 $\pm$ 0.004                          0.31
> > ***
> > The above result is again consistent with our previous findings on the applicability of the IB-GP for analysis of RGCs's responses. We would be happy to add these new results to our supplementary in the updated manuscript.

---

> > > ### Comment · Reviewer_6crN · 2022-08-08
> > > **Acknowledge Response**
> > >
> > > I acknowledge the response of the authors. I appreciate that they have more images then I thought they had and that they improved the presentation. However, the concern about the few neurons largely remains. Also, the image generation still seems oddly regularized and less helpful than one might hope. Thus, I feel my 6 rating still seems quite fitting.

---

> > > > ### Author Response · Authors · 2022-08-08
> > > > **Image generation for regularised and non-regularised cases**
> > > >
> > > > We thank the reviewer for reading through our responses. We would like to take the opportunity to add an explanation about the image synthesis process. Adding the smoothness constraint on the synthesized images has shown greatly beneficial. For example, authors in (Bashivan, science, 2019) have used a 2D total variation constraint directly on the pixel domain of the synthesized images, or in (Walker, Nature neuroscience, 2019) authors use a Gaussian filter on the gradient passed to the images. All this is to prevent the high-frequency noise in the generated images. We also observed the same effect, i.e. the smoothness constraint helped to bring out the optimal pattern out of the noisy background. This observation was valid both for the Brownian dataset as well as the Natural dataset, although more critical for the Natural images. We added a plot in the **supplementary 13** where the image synthesis with and without smoothness constraint are compared. We believe this is due to the fact that the smoothness constraint allows the image synthesiser algorithm to focus on the prominent features in the generated images and to bring the gradient from the RGC responses to the pixels of the synthesized images more easily. As for the application of the image synthesis for the prostheses, specifically sub retinal prostheses that interact with bipolar cells (Farnum 2020), one can imagine that the minimal information optimal stimuli could potentially help with saving energy and resources by stimulating neurones that are contributing to the perception (could be one population or multiple population that are positionally scattered) rather than wasting the limited resources on those that are not. We actually show this for scattered population of neurons in the dataset (although small in number in our case), this is possible. **Please kindly see supplementary 9 and 13**.
> > > >
> > > >
> > > >
> > > > Citations:Farnum A, Pelled G. New vision for visual prostheses. Frontiers in neuroscience. 2020 Feb 18;14:36.

---

### Official Review · Reviewer_G6SJ · 2022-07-11

**Rating:** 6
**Confidence:** 3
**Soundness:** 3 good
**Presentation:** 2 fair
**Contribution:** 2 fair

**Summary:**

The paper proposes a VAE model for generating the ganglion cell responses for visual images. Their motivation is that this could be used for visual prosthetic devices which have a limited number of electrodes so can only handle a sample of the input so it is better to send most important information. They attempt to compress the input images to the lowest possible information required for the response to be identical.

**Questions:**

- I'm not sure I'm understanding this correctly but shouldn't number of latent variable be lower than the number of outputs for this to be a bottleneck model? The latent dimension is swept from 1 to 50 with only 9 neurons
- The IB-GP model appears from figure 3 to learn the order the stimuli were sent in rather than the content of the image
- Line 272 "(equivalent to setting α = ∞ in the 273 objective function of the image synthesizer)". Consider dropping this as it is not mathematically sound. One cannot minimize a function with infinity except turning the factor to negative.
- Can you plot the synthesized images only around the field of view of each neuron? The given synthesized images don't seem to be informational at all to the given original stimuli

**Limitations:**

Given that this was tested on each dataset separately and with training and test sets being from the same distribution, it is highly likely that this model is overfitting to just compress a choice out of 10. The only thing that the model seems to be recognizing well would be that the jittered images are parts of the same category.

**Strengths And Weaknesses:**

Strengths:
- The model appears to outperform the FF CNN model baseline at a larger number of latents.

Weaknesses:
Major:
- I am very skeptical of this model. It appears that the model with temporal information included has learned the sequence of stimuli rather than the actual contents and stored that in one latent variable.
- The minimal information image does not show any resemblance to the original image and it seems that what the model learned is how to create a category of images that maximize the latent distance between categories but does not care about the actual content of the image.
- The above two points can be tested if you add out of distribution categories in your test set.
- The number of latent variables in high performance conditions is larger than the number of RGCs so it is not a bottleneck model

Minor:
Line 78 "the CNN cannot be directly utilized to extract the principal features of the stimuli" this statement is inaccurate, it should be CNN for image classification or something like that, your model is a CNN

---

> ### Author Response · Authors · 2022-08-01
> **Response to the reviewer G6SJ**
>
> # General:
>
> We thank the reviewer G6SJ for reading our manuscript and providing thoughtful feedback. It appears the strongest criticism in the review is about the possibility that the model is overfitting the data and compressing them into 10 categories. However, this assumption appears to be predicated on the misunderstanding that there limited number of category of images in the training and test sets, and the two are very similar. *This is not the case*. We used two dataset in our experiments. One of the dataset is the Natural dataset which consists of images in the wild with no category and labels. We believe the large number of unique images in the dataset and the limited capacity of the network cast away the possibility that our network is just learning the category of images. We will explain this point below in details.
>
> The second concern of the reviewer is about the latent dimension of the IB-GP model being larger than the actual output dimension of RGCs. We will address this point in details below, but we would like to clarify that we varied the number of latents from very low to very high to study how this design choice would affect the the prediction of the neural responses. BUT, we observed that the model is a good predictor of the neuronal responses **even when the latent number is smaller than the output size**.
>
> # Major concerns:
>
> >**1-I am very skeptical of this model. It appears that the model with temporal information included has learned the sequence of stimuli rather than the actual contents and stored that in one latent variable.**
>
> We should clarify that the Natural dataset are from the *Natural images from the birthplace of the human* (Gasper Tkacik et al. 2011). This dataset consists of Natural images in the wild that do not belong to any particular category. Essentially the train set contains 3950 unique images (each unique image is jittered 100 times, so 3950x100 images) and the test set contains 50 unique images (50x100). One batch of test data consists of 10 unique **new** images (10x100=1000). As shown in Figure 3b, the latent correctly identifies this. HOWEVER, this does not mean the model is only learning the sequence and not caring about the content of the images. If the content information of the images were ignored, then the model should not have a good prediction of the RGCs' responses and a similar *latent* autocorrelation as that of the groundtruth neuronal responses since RGCs responses are a nonlinear function of both the spatial and temporal information in the stimuli.
>
>
>
> >**2, 3-The minimal information image does not show any resemblance to the original image and it seems that what the model learned is how to create a category of images that maximize the latent distance between categories but does not care about the actual content of the image.**
>
> We do not agree that the minimally informative images must resemble the original images. Previously published work in the literature that look into finding the optimal stimuli for activating cortical regions obtain complex spatial features that do not resemble the original stimuli (Walker, Nature neuroscience, 2019)(Bashivan, Science 2019). In fact, an optimal stimuli is perhaps one that does not have all the information but only that which is essential for producing the neural activity. The  synthesized images obtained by our model is similarly producing minimal information stimuli that with spatial patterns that maximally activates the neurons. For the Natural dataset, we observed different types of patterns:  locally high frequency features, Gabor-like filters, Gaussian-like patterns, etc. Our findings are consistent with the previous findings in the neuroscience literature that these types of patterns (specially Gabor filters) are those that maximally drive early visual activity (Paninski, Progress in brain research, 2007). The success of the closed-loop algorithm in finding of optimal stimuli is another indicator that the IB-GP model is learning the content as well as temporal dynamics of the data. Continued...

---

> > ### Author Response · Authors · 2022-08-01
> > **continued**
> >
> > We would also like to bring the reviewer's attention to a new experiment we did in which, we used the closed-loop algorithm with the IB-GP model for the Brownian dataset. This dataset is different from the Natural dataset in the sense that it consists of simple images with a constant background of unity (maximum intensity) and disc-like patterns (zero-intensity). This is a simpler dataset than the Natural dataset. We observed that the minimal information stimulation that is generated by the closed-loop algorithm shows more resemblence to the original stimulation. This is due to the simple structure of the original stimulation that is already the close-to-optimal stimulation. We note that the algorithm focuses on the RF regions of the neurons by ignoring the parts of the stimulation that do not contribute to the RGC responses.
> >
> > We believe the results of the above experiment further confirm that the IB-GP model is actually learning the content information of the images. If it were only learning their category, it would not have produced these types of optimal stimuli. We would be happy to add these results in the updated manuscript (please see the supplementary).
> >
> > >**4-The number of latent variables in high performance conditions is larger than the number of RGCs so it is not a bottleneck model**
> >
> > Please see the response to Q1 here.
> >
> > # Minor concerns:
> >
> > Thank you for bring our attention to this. We have changed the wording in our updated manuscript.
> >
> >
> >
> > # Questions
> >
> > >**1- I'm not sure I'm understanding this correctly but shouldn't number of latent variable be lower than the number of outputs for this to be a bottleneck model? The latent dimension is swept from 1 to 50 with only 9 neurons**
> >
> > Initially, we would like to state that in the information bottleneck framework it is standard to use a latent space with sizes larger than the output. For example in (Alemi ICLR 2016), a latent size of 256 is used for the MNSIT classification. We believe the bottleneck has more to do with the structure of the loss function and less to do with its size. Yet, we observed that in fact an IB-GP model whose latent size is smaller than it's output size can well represent the data. we systematically varied the latent size. For the Natural dataset, the latent size of the IB-GP model was varied from 1 to 15. For the Brownian dataset, the latent size was varied from 1 to 50. In both cases, as shown in Figure 2, the model can obtain a similar to Feedforward CNN prediction accuracy with when the latent sizes are 4 (Natural dataset) and 8 (Brownian) which are smaller than output size of 9 in both cases. Moreover, even when the latent size is larger than the output size, for example a latent size of 15 for the Natural dataset, the ablation analysis shown in Figure 3a shows that one latent is enough to have both a good neural prediction of the output responses as well as informative representation of the latent dynamics of the responses as demonstrated by the matching autocorrelation of latent dynamics and RGCs' dynamics.
> > This ablation analysis is a standard practice in VAE models where initially larger latent sizes are used but later the latents are ablated based on the amount of informative knowledge picked up by each latent (Higgins et al. 2016).
> >
> > >**2- The IB-GP model appears from figure 3 to learn the order the stimuli were sent in rather than the content of the image**
> >
> > Please see the responses to the Major concerns 1-3.
> >
> > >**3- Line 272 "(equivalent to setting α = ∞ in the 273 objective function of the image synthesizer)". Consider dropping this as it is not mathematically sound. One cannot minimize a function with infinity except turning the factor to negative.**
> >
> > Thank you for this point. We have modified the manuscript.
> >
> > >**4- Can you plot the synthesized images only around the field of view of each neuron? The given synthesized images don't seem to be informational at all to the given original stimuli**
> >
> > This a good point. We have modified Figure 4 to better visualize the synthesized stimuli around the RF of neurons. We have also added a few more examples of the optimal stimuli in the supplementary.
> >
> > # Limitations
> > Please see the responses above.

---

> > > ### Comment · Reviewer_G6SJ · 2022-08-04
> > > **Much clearer**
> > >
> > > Thank you very much for the response. I understand the dataset much better now. I think it is still a bit ambiguous in the paper so maybe add this explanation to the main text as it seems I was not the only one who had this skepticism.
> > >
> > > The result 9 figure 3 a & b and paragraph at page 6 line 244 is still very strange and requires an explanation of the following:
> > > - Why is it that almost all the information of natural images are centered around one latent dimension? This is especially strange as in figure 2 it shows the addition of more latent dimensions gradually improved performance which is not-consistent with this amount of information centralization.
> > > - The clustering of the 10 categories in figure 3b. Since you mention that the test set has 50 images, why did you only present one batch of 10? I suggest showing the whole test dataset.
> > >
> > > I also suggest showing the same metrics for the training dataset as a possible way of understanding this discrepancy.

---

> > > > ### Author Response · Authors · 2022-08-05
> > > > **Response to the Reviewer**
> > > >
> > > > We thank the reviewer for reading the responses. We are pleased that the confusion about the dataset and our model is cleared up. At the request of the reviewer, we added the explanation about the construct of the dataset to the manuscript. Now, we will address the remaining questions and hope our responses will clear up the remaining concerns.
> > > >
> > > > > **Why is it that almost all the information of natural images are centered around one latent dimension? This is especially strange as in figure 2 it shows the addition of more latent dimensions gradually improved performance which is not-consistent with this amount of information centralization.**
> > > >
> > > > This is a good observation. We believe that the reason behind this is that the most informative latent in the model learns the temporal dynamics elicited by the stimuli, such as the onset of spikes and the time lags, while the rest of the latents learn the exact firing related to the luminance, etc. This can be tested by doing a traversal analysis as well as an ablation test on the latents. The traversal analysis was already reported in the supplementary S8. It can be seen that traversing the second most informative latent only changes the firing rates rather than the onset of the spikes.
> > > >
> > > > The ablation test, on the other hand, reveals that more than one latent is required to predict the exact rates.
> > > > The test is as follows. Initially, the latent variables are all zero-masked except for the most informative variable. This ablated model is then tested on the test dataset to produce the RGC predictions. Next, the two most informative latents are kept, and so on. The Pearson correlation in each case is reported in the table below.
> > > >
> > > > **Number of latents kept**     1         2         3          4            5  ....  15
> > > > ***
> > > > **Pearson correlation**        0.37   0.40     0.43      0.43     0.45     0.47
> > > >
> > > > This shows that the added information stored in the learned higher order latents gradually improves the accuracy. This is consistent with the results in Fig 2. Therefore, as it can be seen in the table above, the centralised info in one-latent ablated model is sufficient at least for the latent dynamics (Fig3b), adding extra latents can help with improving the neural prediction of the model.
> > > >
> > > > > **The clustering of the 10 categories in figure 3b. Since you mention that the test set has 50 images, why did you only present one batch of 10? I suggest showing the whole test dataset.**
> > > >
> > > > At the suggestion of the reviewer, we now plot the low-dimensional representation of the latent space for the entire test set. Again, the test set contains 50 unique images that are grouped in batches of the size 10. The results are presented in Fig 3b (IB-GP) and 3e (IB-Disjoint). Each batch is represented by one color (10 clusters in one color). It can be seen that the IB-GP can distinguish the dynamics of each unique image (1x100 jittered).
> > > >
> > > > > **I also suggest showing the same metrics for the training dataset as a possible way of understanding this discrepancy.**
> > > >
> > > > This is a good point. We added the plots showing the metrics for the train set. Please see supplementary 12. As it can be seen, the train and test metrics follow the same trend. It can be seen in the plots that the IB-GP model does not overfit the data. One observation here is that the train set requires a few more latents before the curves flatten. This is due to the fact that the train set's RGC responses are count data (integer numbers), whereas the test set's RGC responses are averaged responses of repeated numbers. This is the standard practice (McIntosh et al. 2016). The former requires more latents to fit the data. Also, the Pearson correlation of the train set is smaller than the Pearson correlation of the test set. This is standard for every model for count data versus firing rate. We would be happy to add this explanation to the updated manuscript.
> > > >
> > > > Citation:
> > > > L. McIntosh, N. Maheswaranathan, A. Nayebi, S. Ganguli, and S. Baccus. Deep learning models of the retinal response to natural scenes. Advances in neural information processing systems, 29:1369–1377, 2016.

---

> > > > > ### Comment · Reviewer_G6SJ · 2022-08-07
> > > > > **Thank you**
> > > > >
> > > > > Thanks a lot for the improvements and responses. I believe this is much clearer and have increased the score based on this.

---

> > > > > > ### Author Response · Authors · 2022-08-07
> > > > > > **Thank you**
> > > > > >
> > > > > > We would like to thank the reviewer for reading through our responses and providing feedback that helped to improve our manuscript.

---

### Official Review · Reviewer_SLVr · 2022-07-13

**Rating:** 7
**Confidence:** 4
**Soundness:** 3 good
**Presentation:** 2 fair
**Contribution:** 3 good

**Summary:**

The authors propose to use the Information Bottleneck method to compress the visual input while capturing a low-dimensional dynamic of the Retinal Ganglion Cell (RGC) spike trains.
Many variations of their model and one other CNN-based model are compared. Performances reach similar levels but one of their model is performing best both in term of Pearson Correlation and Negative Log-Likelihood.
The authors also propose to generate images which would produce a similar RGCs activity while reducing their "complexity".

**Questions:**

- image complexity is not quantified (figure 4),
- I do not understand why the latent variable Z must have high mutual information with Y and low mutual information with X ? Can you give some intuition about how it achieve its goal ?
- In other works, it is sometimes assumed that mutual information between X and Y is maximized. How does it relate to information bottleneck method ?
- Its seems the latent Z has mainly learned stimulus labels, I don't understand why it is presented as an inferred dynamic ? (Figure 3 be)
- What is the interest of the closed-loop image synthesis ? Do you obtain a "population RF" ?
- In figure 4, we can only see the RF location not the actual filter (or maybe it's too small). If the obtained image is some sort of population RF, is linearly predictable from the individual RF obtained by STA ?


Remark :
- Eq (7), if it's a lower bound do not write I_{IB} \approx  but I_{IB} >=.


Minor remarks :
- l 106 (and where applying) : do not start a sentence with a mathematical variable,
- use \cite{smith2020blabla,tommy2021bliblou} to get [1,2] instead of [1][2],
- the text in all figures is too small,
-

**Limitations:**

What is the energy consumption of the training ? How many GPUs ? For how long ?
Does the gain in performance compared to non-CNN/DNN models justified in comparison to the training cost (duration, consumption, CO2) ?

**Strengths And Weaknesses:**

Strengths :
- the model is sufficiently described
- good performances
- model comparison

Weaknesses :
- comparison to a single model from the literature (are there non-CNN/DNN based method out there ?)
- the learned dynamics are not sufficiently shown, I would expect to see something like a trajectory as function of time
- the closed-loop image synthesis is not sufficiently motivated. There is a lack of intuition behind the resulting images.

---

> ### Author Response · Authors · 2022-08-01
> **Response to the reviewer SLVr**
>
> # General
> We thank the reviewer SLVr for taking the time to review our manuscript. We appreciate the positive feedback and the constructive comments. We hope we can satisfactorily address the raised questions/comments in our responses. It appears the strongest criticism in the review is about the intuition behind the closed-loop image generation process and if the resulting optimal stimuli are RF population. The optimal stimuli are not population RF. We will address this point in detail below.
>
> Besides this, the second concern of the reviewer relates to the dynamics of the latents and the possibility that they are learning only the stimulus labels. We will address this point in detail below but before that we want to clarify that the data the models are trained on do not have any particular labels or annotations. The two sets of data in our experiments consist of dynamic stimuli and the sequence of responses. Additionally, the number of unique stimuli in the dataset is large enough to cast away the possibility of the model overfitting any types of categories.
>
> # Major Concerns
> >**1-comparison to a single model from the literature (are there non-CNN/DNN based method out there ?)**
>
> There are a number of non-CNN model in the literature such as Generalized Linear Models. Although they can successfully encode artificial stimuli to RGC responses, they fall short of faithfully describing RGCs responses to natural stimuli (McIntosh et al, 2016). Currently, feedforward CNNs are the state-of-the-art for RGC responses. Therefore, we compared the results only to CNNs.
>
> >**2- the learned dynamics are not sufficiently shown, I would expect to see something like a trajectory as function of time**
>
> Please see the response to Q4 here.
>
> **3- The closed-loop image synthesis is not sufficiently motivated. There is a lack of intuition behind the resulting images.**
>
> Previously published work in the literature have looked into closed-loop image synthesis, particularly to find optimal stimuli that maximally activate neurons in the Retina (Paninski, Progress in brain research, 2007), or more recently in V1 and V4 regions in the mouse (Walker, Nature neuroscience, 2019)(Bashivan, science, 2019). The goal is to obtain spatial features that not necessarily resemble the original stimuli but still elicit the same responses. This allows to correctly use the resources available, for example by placing the stimulation energy at the right locations and with the correct spatiotemporally features. Therefore, using techniques that focus only on obtaining the RF locations of neurons such as STA, etc. is perhaps not the optimal strategy. The IB-GP synthesized images obtained by our model is doing the same task. For the Natural dataset, we observed different types of patterns: locally high frequency features, Gabor-like filters Gaussian-like patterns, etc. Our findings are consistent with the previous findings in the neuroscience literature that these types of patterns are those that maximally drive early visual activity. We would be very happy to add this discussion to the supplementary of the updated manuscript.
>
> # Questions
>
> >**1: Image complexity is not quantified (figure 4)**
>
> This is a good point. We agree that the complexity is not well quantified. But, we used the 2D Total Variation loss as a metric to somehow represent the complexity of the synthesized and original images. Perhaps, it is also useful to use the entropy of images where entropy is defined as $-\sum_{i}^{} p_i\log_2{p_i}$ with $p_i$ being the histogram of the images. As expected, we observed that the entropy of the synthesized images are smaller than that of the original images. We would be happy to include these results in the updated manuscript.
>
> >**2 and 3:  Relations among X, Y and Z and their mutual information**
>
> The input images X produce output RGC responses Y. We then ask this question *if all information in X is needed to obtain Y*? This is because the channel (retinal circuit from X to Y) might be lossy and/or Y is essentially a subset of responses of all possible neurons. Using the IB framework, we constraint the latent Z to be minimally informative about X. On the other hand, we want to be able to produce Y as much as possible. So the mutual information between Z and Y needs to be maximized. Therefore, the IB framework is essentially a way of maximizing the mutual information between X and Y, but through a proxy, i.e. latent Z.

---

> > ### Author Response · Authors · 2022-08-01
> > **Continued**
> >
> >
> > >**4: Its seems the latent Z has mainly learned stimulus labels, I don't understand why it is presented as an inferred dynamic ? (Figure 3 be)**
> >
> > We do not agree with this statement that the latent has only learned the stimuli labels because one: the dataset has no label information. The Natural stimuli dataset are from the *Natural images from the birthplace of the human* (Gasper Tkacik et al. 2011). The images do not belong into any particular category. Two, no label information was provided to the latent externally. Therefore, the latent must have learned the content information of the images. Accordingly, the TSNE analysis of the latent in Figure 3b,e for one sample of the test set (10 **new** unique images each jittered for 100 times) confirms this fact (10 clusters in Fig 3b). At the same time, since the jittered images elicit similar dynamics at the RGCs level, their low dimensional representation should also be similar (jittered images are in the same cluster). We agree with the reviewer that Fig. 3b,e alone, do not fully confirm if the dynamics of the RGCs responses is represented in the latent space. We will change our wording and address Fig 3b,e differently. BUT, what can confirm the latter is the plot in Figure 3c. It shows the Auto Correlation (AC) of the groundtruth, the averaged model output and the latent as a function of time lags. The AC of latent closely mimics that of the true responses. Similar plots for each neurons in the dataset were provided in the supplementary 2. Also, we studied the learned covariance matrices of the model as a function of time and the traversal analysis of the latents to further understand the dynamics of the latent. This results were given in supplementary 8.
> >
> > >**5, 6: What is the interest of the closed-loop image synthesis ? Do you obtain a "population RF" ? In figure 4, we can only see the RF location ...?**
> >
> > Please see the response to Q1 in the reviews of Reviewer 6crN for more details about the closed-loop procedure. The purpose of the image synthesis is not to obtain a population RF but to find a spatiotemporal pattern that maximally activates the neurons. These patterns, of course happen to be within the RF region of the neurons. But what actually is activating them maximally is the structure of the stimulation in terms of regions of high intensity, sharp edges, etc and their temporal footprint. Since the RGCs of the neurons are nonlinear, linear STA cannot produce these stimuli. At the request of the reviewer we have updated Fig 4 and added more examples of optimal stimuli in the Appendix.
> >
> > # Limitations
> >
> > > **What is the energy consumption of the training?...**
> >
> > The IB-GP model was trained on a single Nvidia RTX 3090 GPU which takes about one hour of compute time to train. The Feedforward CNN would take about 20 mins of compute time. We believe the compute time of IB-GP model is still reasonably short to justify its usage

---

> > > ### Author Response · Authors · 2022-08-09
> > > **Author-reviewer discussion deadline**
> > >
> > > Dear reviewer. Just a kind reminder that the author-reviewer discussion ends very soon. We would love to have your feedback on our responses as we believe there were some points which we were intially not so clear about but we tried to clear up the confusion using your constructive feedback. We indeed believe your comments helped us to improve our manuscript and want to have a chance to clear up the remaining confusion, if any. Thank you very much for your time spending on this manuscript.

---

> > > > ### Comment · Reviewer_SLVr · 2022-08-09
> > > > **Thanks for the feedback**
> > > >
> > > > You've answered to some of my concerns and I have raised my score. Yet, the competition is tough and I encourage the authors to pursue their efforts whatever the outcome of the review process is. Maybe, it would help to have more convincing synthesis results regarding the ambitious goal of "bring[ing] the vision to the blind".

---

> > > > > ### Author Response · Authors · 2022-08-09
> > > > > **Thank you**
> > > > >
> > > > > We thank the reviewer for the comments/feedbacks and are happy that we could answer some of the reviewer's concerns. We believe our work may not be an immediate solution to the prostheses problem but it is a step forward.
> > > > >
> > > > > We thank the reviewer for suggesting to look more into the synthesis problem. We want to state that the synthesised images provide the same responses as those of the Natural images. Also their structure, various Gabor-like filters, on- and off- Gaussians, etc., is consistent with the previous knowledge in the literature. In the future iteration of the work, we will look more into these. For example, it is interesting to see how the minimal information stimuli and the constrained latent dimension of the IB-GP (which is tuneable via the beta parameter in the model) is related to the achieved perception rather than the spiking activity of the neurons. For all these reasons, we believe our model could be interesting for the neuroscience community.
> > > > >
> > > > > We revised the Abstract of the manuscript to summarise the findings of this work.

---

### Official Review · Reviewer_daMk · 2022-07-16

**Rating:** 5
**Confidence:** 2
**Soundness:** 3 good
**Presentation:** 2 fair
**Contribution:** 3 good

**Summary:**

The paper reports the results of analysis and simulation using the information bottleneck method to find a space of reduced stimuli that produce the same population neural response as complex natural stimuli.


**Questions:**

Figure 3 shows IB-GP and IB-Disjoint, but which is which in the figure?  also what do the blue vs. grey traces correspond to in panels g,h?


**Limitations:**

see above

**Strengths And Weaknesses:**

Strengths
- a principled approach using IB method

Weaknesses:
- motivation is not clear, difficult to ascertain the punchline of what has been achieved here and how it helps us to understand the retina or build a better prosthesis.

The work appears to be motivated by the desire to build more effective neural prostheses for vision.  Current technology can only stimulate a fraction of neurons in a region of retina, resulting in un-natural percepts.  The authors then turn to the question of whether you can derive a reduced stimulus space that yields the same neural responses as complex natural images.  This seems like a non-sequitur.  If you want to fix the neural prostheses problem, wouldn't you want to find a modified stimulation protocol that yields more natural percepts?  This would presumably require identifying precisely which neurons you are stimulating and somehow adjusting the stimulation so that the cortex can fill in the desired percept from the fraction of neurons you are stimulating.  Instead the authors turn to the question of deriving a simplified stimulus space.  Although a valid scientific question, I don't see how it addresses the neural prostheses problem.  Also, by the end of the paper, I don't see how it helps us to understand retinal mechanisms in general.  If you are only recording from a fraction of neurons then of course there will be a null space (not necessarily linear) in the stimulus space.  Even if you had access to an entire population, there would still be a null space because the retinal transform is certainly lossy.  One could hypothesize how lossy based on our current understanding of retinal mechanisms and compare this to results of stimulating with derived stimuli as shown here, but that's not something this paper does.  So I'm left scratching my head.  Overall it seems like a very nice theoretical and principled approach, but the scientific question and what we learn from this is missing.

---

> ### Author Response · Authors · 2022-08-01
> **Response to the reviewer daMK**
>
> # General
>
> We thank Review daMk for taking the time to review our manuscript. It appears the strongest criticism in the review is a confusion about the scientific question this manuscript is trying to answer and how this manuscript contributes to that. We will explain the points made by the reviewer in detail later but we would like to state that although our proposed method may not be an immediate solution for the prostheses, we believe it is indeed a step forward. We support the approach taken in this manuscript using the published work in the literature that are closely aligned with our work here. Therefore we feel this confusion should not be a strong justification to recommend rejection. As we pointed out in the manuscript, we want to clarify that the contributions of this manuscript are 1- proposing a novel systematic approach for modelling natural scene responses of the RGCs using information bottleneck (IB) in which we introduce the IB-GP so that it can be applied to dynamic neuronal responses. To the best of our knowledge, except for the CNN based models of McIntosh et al., RNN model of Batty et al. and similar follow-upped works, no other ML based models is proposed. 2- We not only show that it is a good output response but dynamics is well captured.
>
> # Major Concerns
>
> > **This seems like a non-sequitur. If you want to fix the neural prostheses problem, wouldn't you want to find a modified stimulation protocol that yields more natural percepts?**
>
> We disagree that the efforts to improve the prostheses problem should only be focused on direct stimulation protocols. We would like to refer to this seminal work (Nirenberg et al. 2012) that shows a better encoder of the retinal responses could increase the prosthetics capabilities far beyond other strategies such as increasing the number of electrodes, etc. Therein the authors build an encoder of the RGC responses and use the encoder's predicted neural code with a prosthesis  to stimulate neurons. This is exactly the approach presented in our manuscript, i.e. building a principled good encoder model of the RGCs responses.
>
> Citation:
> S. Nirenberg et al., Retinal prosthetic strategy with the capacity to restore normal vision. PNAS, 109(37):15012–15017, 2012.
>
>
> > **Instead the authors turn to the question of deriving a simplified stimulus space. Although a valid scientific question, I don't see how it addresses the neural prostheses problem.**
>
> We believe that finding the minimal information stimulus space that elicits the same neuronal responses could potentially help with finding a stimulation protocol that allocate the energy and resources better so as to obtain a better percept.
>
>  > **Also, by the end of the paper, I don't see how it helps us to understand retinal mechanisms in general. If you are only recording from a fraction of neurons then of course there will be a null space (not necessarily linear) in the stimulus space.**
>
>  We would like to clarify that minimal information algorithm does not only find null space of the stimulation. On the contrary, it finds the minimum information stimulation that maximizes the neurons' activities. As shown in Figure 4, these patterns are spatiotemporal patterns with specific spatial features. This method is based on previous work (Paninski, Progress in brain research, 2007), or more recently in V1 and V4 regions in the mouse (Walker, Nature neuroscience, 2019)(Bashivan, science, 2019).
>
>  We observed that the minimum stimulus consist of different types of patterns: locally high frequency features, Gabor-like filters Gaussian-like patterns, etc. Our findings are consistent with the previous findings in the neuroscience literature that these types of patterns are those that maximally drive early visual activity. We would be very happy to add this discussion to the updated manuscript.
>
> > **Overall it seems like a very nice theoretical and principled approach, but the scientific question and what we learn from this is missing.**
>
>   We appreciate that the reviewer believes this work is theoretically solid and a principled approach. We believe we can clarify any confusion about the scientific goals of this work easily in the present manuscript. Accordingly, we revised the Abstract and the introduction.
>
> # Questions
>
> >1. **Figure 3 shows IB-GP and IB-Disjoint, but which is which in the figure? also what do the blue vs. grey traces correspond to in panels g,h?**
>
> The first row in Figure 3 plots the learned dynamics of the IB-GP model and the second row shows the same dynamics learned by the IB-Disjoint. We would be happy to add tags to the figure to show each case more clearly.
>
> The green curves in the Figure 3(g) plot the *predicted* neuronal responses obtained by the IB-GP whereas the grey curves are the ground truth responses. Figure 3(h) plots the same curve using the IB-Disjoint model.

---

> > ### Author Response · Authors · 2022-08-08
> > **Update to the manuscript**
> >
> > We want to bring the reviewer’s attention to the new result in Supplementary 9, 13 where we visualized the synthesized images from the Natural and Brownian datasets. For the former, the available RF population in the dataset is mostly centered, while for the latter, the RF population is scattered along the image coordinate. We can see that the image synthesizer automatically identifies precisely which neurons are contributing to the RGCs spikings and finds the minimal information stimulation that maximally activates them using the information learned by the IB-GP model.
> >
> > We believe the applicability of the IB-GP model and the image synthesis algorithm for prostheses is important in two aspects. One, as the IB-GP model is a good encoder of the Retinal neural codes, it can be used directly with a transducer, as done in Nirenberg 2012, to stimulate RGCs directly. Second, the minimal information stimuli could be used directly with sub-retinal prostheses and bipolar cells. This minimal stimulus is important as it could save resources by avoiding stimulating cells that do not contribute to the perception and importantly, activating them with minimal patterns that are obtained using the image synthesizer algorithm.
> >
> > Generally, we believe our model could help understand the information processing in the visual system and the brain. For example, in the future iteration of the work, it is interesting to see how the minimal information stimuli and the constrained latent dimension of the IB-GP (which is tuneable via the beta parameter in the model) is related to the achieved perception rather than the spiking activity of the neurons. Therefore, we believe our model could be interesting for the neuroscience community.

---

> > > ### Author Response · Authors · 2022-08-09
> > > **Author-reviewer discussion deadline approaching**
> > >
> > > Dear reviewer. Just a kind reminder that the author-reviewer discussion ends very soon. We would love to have your feedback on our responses as we believe initially there might have been some confusions but we tried to clear them up using your feedback. We believe your comments helped us to do so and want to have a chance to clear up the remaining confusion, if any. Thank you very much for your time spending on this manuscript.

---

### Author Response · Authors · 2022-08-01
**General response**

We would like to thank the reviewers for taking the time to read our manuscript and providing their feedback and thoughtful raised questions/comments. At a high-level, our response to the reviewers is focused on clarifying the scientific questions this manuscript is investigating: 1- if the IB-GP model is able to learn a low dimensional representation of the dynamic stimuli and neuronal responses and 2- the rational behind the closed-loop image synthesis. We will respond to the misunderstanding that has raised about the IB-GP model and describe additional experiments we ran to further validate it.

We want to clarify that one of the main contribution of the IB-GP is its novel and principled approach for representing the neural activity of the RGCs with a low dimensional embedding. Not only our model does perform competetive to the state-of-the-art baseline for RGCs neural activity, but it is used to obtain simple stimuli/filters that derive neurons similarly as derived by complex high-dimensional stimuli. We believe that the close similarity between the autocorrelation of the RGCs' responses and autocorrelation of the latent space in different time lags, as well as the traversal analysis of the latent space confirm that the IB-GP model learns low dimensional representations of neuronal responses.

---

### Meta-Review · Area_Chair_AtPd · 2022-08-24

**Recommendation:** Accept
**Confidence:** Certain

**Metareview:**

In this paper, the authors describe a new method for estimating the stimulus-response characteristics of biological visual neurons from the retina. The authors employ the Information Bottleneck method to compress the visual representation and compare their results to other models including CNN-based architectures. The authors find that their model is most performant in terms of Pearson correlation and log-likelihood on real neural spike trains recorded in response to natural imagery and Brownian movement. Lastly, the authors use the model to reconstruct stimuli using the learned latent representation.

The reviewers applauded the principled approach to the estimation and analysis of neural receptive fields, the clarity of presentation of the method and comparisons across models and the scale of the data. In the responses, the authors were able to showcase new results indicating that the method could scale to a larger number of neurons. One reviewer did take issue with the introduction relying too heavily on the discussion of neural prosthetics to motivate this work as opposed to a discussion of just the neural coding problem. I agree with the reviewer in this sentiment and would like to see the introduction of the paper updated accordingly to better emphasize that this work is focused on the issue of modeling the stimulus-response relationship of a neural population.

Given the strong support of the reviewers, this paper will be conditionally accepted provided two updates are performed by the authors: (1) update to the introduction section and (2) update to Figure 4D to fit within the margin.


**Award:**

No

---

### Decision · Program_Chairs · 2022-09-14

Accept